# ROSETTA: CONSTRUCTING CODE-BASED REWARD FROM UNCONSTRAINED LANGUAGE PREFERENCE

**Sanjana Srivastava\*[1,2], Kangrui Wang\*[1,2], Yung-Chieh Chan\*[1,4], Tianyuan Dai[1,5], Manling Li[1,3], Ruohan Zhang[1,3], Mengdi Xu[1,6], Jiajun Wu[1], & Li Fei-Fei[1,7]**

[1]Department of Computer Science, Stanford University
[2]Together AI
[3]Department of Computer Science, Northwestern University
[4]Scale AI
[5]Department of Computer Science, University of Texas at Austin
[6]Institute for Interdisciplinary Information Sciences, Tsinghua University
[7]Human-Centered Artificial Intelligence Institute
Correspondence: `ssrivastava@together.ai`

## ABSTRACT

Intelligent embodied agents not only need to accomplish preset tasks, but also learn to align with individual human needs and preferences. Extracting reward signals from human language preferences allows an embodied agent to adapt through reinforcement learning. However, human language preferences are unconstrained, diverse, and dynamic, making constructing learnable reward from them a major challenge. We present ROSETTA, a framework that uses foundation models to ground and disambiguate unconstrained natural language preference, construct multi-stage reward functions, and implement them with code generation. Unlike prior works requiring extensive offline training to get general reward models or fine-grained correction on a single task, ROSETTA allows agents to adapt online to preference that evolves and is diverse in language and content. We test ROSETTA on both short-horizon and long-horizon manipulation tasks and conduct extensive human evaluation, finding that ROSETTA outperforms SOTA baselines and achieves 87% average success rate and 86% human satisfaction across 116 preferences.

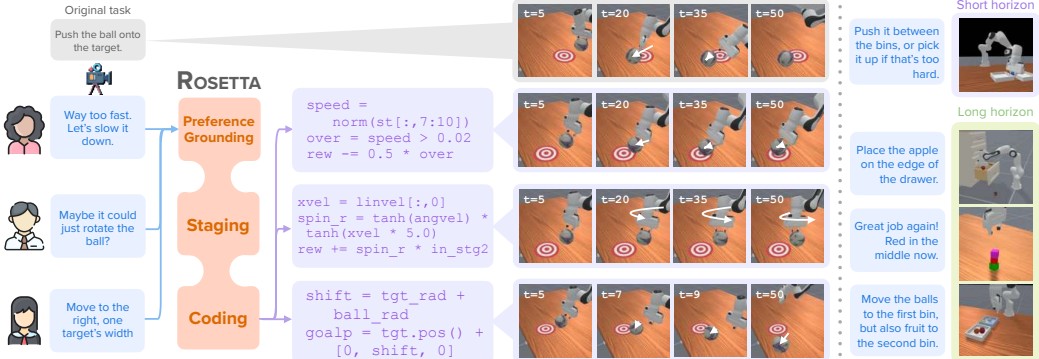

Figure 1: ROSETTA's three-module structure takes unconstrained, unseen natural language preference and generates reward code that incentivizes desired behavior. ROSETTA can adapt to diverse behavioral preferences, not only in position but orientation and speed. We validate ROSETTA's performance in two short-horizon and three long-horizon task-agnostic environments.

# 1 INTRODUCTION

Human-centered embodied intelligence requires that humans be able to guide embodied agents to align with their preferences (Bobu et al. (2024); Leike et al. (2018)). For agents operating in closed-loop interactions, this means aligning with each human preference in its most natural and unconstrained forms (Li et al. (2022)), as shown in Fig. 1. Reinforcement learning (RL)-based embodied agents have demonstrated the ability to adapt to high-level tasks originating in human language given dense rewards (Srivastava et al. (2021)), and the efficiency of specifying reward rather than collecting extensive training data makes RL a promising testbed. However, humans have unique voices and changing goals. Adaptation requires handling unconstrained language and unseen goals that edit, build on, or even contradict prior goals at every step. Generating effective rewards under such conditions is an open problem.

Reward modeling enables creation of nuanced reward signals without expert shaping. While many existing methods are general and multi-task, often able to adapt to changing, unseen goals, they require extensive end-to-end training (Chen et al. (2021); Arora & Doshi (2020)). Due to their large-scale training across diverse domains, foundation models (Bommasani et al. (2022)) offer a compelling alternative to custom reward model training. Their code generation capabilities offer a natural conduit between language and dense scalar reward. However, prior methods like Eureka (Ma et al. (2024b)) and Text2Reward (Xie et al. (2024b)) that work for embodied agent manipulation are limited to clean and structured language, predetermined goals that remain fixed over the course of interaction with the human, and several rounds of feedback that aims only to refine that fixed goal, rather than assert the human's own evolving preferences and receive immediate adaptation.

In this paper, we aim to generate rewards for language preferences that do not obey these constraints. First, **the underlying tasks requested in these preferences are not cleanly phrased**. Individuals express themselves in whatever way is natural and efficient. Second, **the preferences have dynamic and unseen content that requires single-step adaptation**. This means that whatever goal comes up, we aim for a good reward function in a single step. And third, **evaluation of adaptation to naturally expressed preferences is itself a bottleneck**. It is impossible to measure alignment with training metrics such as success rates alone, as they do not address adherence to the preference, only to the goal in the reward function which may not match the preference.

To address these challenges, we present ROSETTA (Fig. 2): **R**eward **O**bjectives from **S**pontaneous **E**xpression **T**ranslated **t**o **A**gents. ROSETTA is a code-based reward generation pipeline that enables embodied agent adaptation to human natural language preference in a single step. ROSETTA generates rewards from preference statements that have no constraints on language, limited constraints on content, and compound over time in long interactions. It contains a **Preference Grounding** module to disambiguate and contextualize in prior preferences and execution, as well as a **Staging** module that repurposes LLMs' known planning abilities (Huang et al. (2022a)) to add structure and stage rewards. Both use `gpt-4o` (OpenAI et al. (2024a)).The **Coding** module then turns the staged reward specification into coded reward function with `o1-mini` (OpenAI et al. (2024b)).ROSETTA therefore generates rewards even from preferences that change the entire shape of a task in just a few words.

We also propose an evaluation framework consisting of three key metric categories. Most important is **alignment**, measuring human satisfaction with the policy. Allowing unconstrained preferences means that unlike in prior work, training metrics only indicate how well the RL algorithm optimized the reward, not whether the reward itself is aligned with human intent. Measuring alignment requires direct human input (Conitzer et al. (2024); Delgado et al. (2023)), but can be noisy and incomplete, motivating additional metrics. First, we do also want rewards that are suitable for RL. We therefore measure reward **optimizability** via policy success rate. Second, looking only at the trained policy is insufficient for determining whether the reward reflects the preference. It may match semantics but not be optimizable, policy rollouts given to the human have some stochasticity, and the human may rate a behavior well because they like it even though they didn't ask for it. We therefore directly measure **semantic match**, measured by expert evaluation of the reward code itself.

We validate ROSETTA iteratively: multiple steps of taking a language preference, generating a reward, training an agent, evaluating, and taking a new preference. We evaluate on 35 sequences of two to four preferences each, total 116 preferences, in five task-agnostic manipulation environments (Fig. 1). ROSETTA successfully interprets ambiguous language, adapts to unseen preferences even after four interaction steps, and produces semantically matched and optimizable rewards that result in aligned

policies. It outperforms Eureka and Text2Reward (Sec. 5.2), demonstrating the importance of its structured approach for immediate and general adaptation. We present three main contributions:

- ROSETTA, a framework that generates code-based rewards. These enable embodied agents to adapt to unconstrained, diverse, dynamic human language preferences online.

- An evaluation framework for measuring human-embodied agent interactions in the wild, consisting of metrics for alignment, semantic match and optimizability.

- Evaluation in five manipulation tasks on 116 human preferences, showing ROSETTA's state-of-the-art performances: 86% human satisfaction, 78% semantic match, and 87% success rate. It shows 33% better human satisfaction than baselines.

## 2 RELATED WORK

**Foundation models on human-robot interaction and control.** foundation models have transformed the landscape of robotics by enabling natural language interaction. Early works focused on affordances and executable control code (Ahn et al. (2022); Liang et al. (2022); Huang et al. (2022b)), establishing the foundation for language-guided robotics. Recent work has explored more interactive paradigms, developing systems that can interpret and incorporate real-time verbal feedback and corrections during task execution (Cui et al. (2023); Liu et al. (2023); Zha et al. (2023); Shi et al. (2024; 2025)). These interactive approaches have been further extended to handle complex spatial reasoning and hierarchical task planning (Huang et al. (2023); Li et al. (2023); Wang et al. (2023); Chen et al. (2023)). The field has also seen significant advances in open-ended embodied control and personalized human-robot interaction systems (Wu et al. (2023a;b); Li et al. (2024b)).

**Foundation models in reinforcement learning.** The emergence of foundation models has sparked new directions in reinforcement learning, particularly in the context of robotic tasks. Vision-language models (VLMs) have proven effective as both success detectors and zero-shot reward generators (Du et al. (2023); Wang et al. (2024)). Developments in value learning have explored personalization of multi-objective rewards and in-context value function adaptation (Hwang et al. (2023); Ma et al. (2024a)), capturing nuanced human preferences.

**LLMs for reward engineering.** The most relevant developments to our work lie in the direct application of LLMs for reward engineering. Recent approaches have demonstrated increasingly sophisticated methods for translating natural language into reward functions, ranging from basic reward shaping to full code generation (Xie et al. (2024a); Ma et al. (2024b); Yu et al. (2023); Li et al. (2024c); Clark et al. (2025); Yu et al. (2025)). Some approaches have explored automated reward generation through progress functions (Sarukkai et al. (2024)) or evolutionary techniques guided by human feedback (Hazra et al. (2024)). These methods require a fixed task definition. They also require pre- and hand-defined goal-specific supervision like fitness functions or goal-specific information in the prompt.

Existing approaches require either pretraining or fixing the goal and using only feedback that is clean and helps achieve that goal. Though ROSETTA is a prompting framework like Eureka and Text2Reward, its structured, domain knowledge-informed reward generation enable adaptation to unseen, unconstrained, dynamic goals without training. The core innovations lie in the problem setting of natural human preferences without constraint on preference; the learning task set up by our data; the conceptual structure underlying our approach; and the evaluation suite.

## 3 PROBLEM FORMULATION

Our aim is to turn human language preferences into semantically matched, optimizable reward functions that lead to aligned embodied agent policies. We aim to do so for ongoing chains of preferences, adapting to each individual preference while also adapting to the compounding meaning over time and avoiding degradation in later interactions. We formulate it as the reward generation problem for unconstrained human language preference adaptation, extending the reward generation problem presented in Ma et al. (2024b).

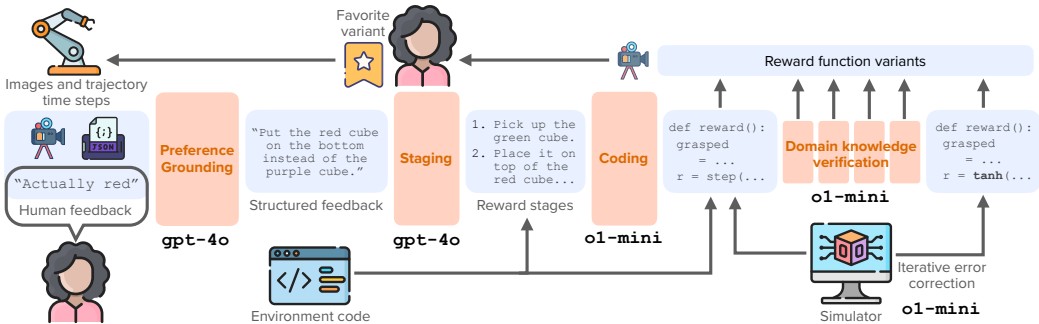

Figure 2: ROSETTA diagram. A trajectory from the latest policy and preference history are given to the Preference Grounding module. The resulting grounded preference is given to the Staging module. The Coding module takes resulting staged reward plan and generates reward variants with various domain knowledge reasserted. Iterative error correction conducted.

The reward generation problem (Ma et al. (2024b)) takes a task description $l$ and generates a reward function $R$ that maximizes a known fitness function $\bar{F}(\mathcal{A}_M(R))$, where $\mathcal{A}_M(R)$ is a learning algorithm run on world model $M$ and reward $R$ to get policy $\pi$. The fitness function $\bar{F}$ is known when the task goal is predetermined and does not change. With unconstrained human preferences, the fitness function denoted as $F^{(t)}$ is instead unknown and time-dependent. It can be used to aid the adaptation process across different $t$, but unlike in Eureka or Text2Reward, we do not assume this. Instead, we aim for each $F^{(t)}$ to be maximized, so that we minimize user queries for any one goal. We note that the fitness function exists only in the mind of the annotator and their satisfaction is the objective of the method, meaning human involvement is unavoidable.

**Reward generation for unconstrained human language preference adaptation.** An adaptive embodied agent interacts with humans through multiple iterations of preference. At preference iteration $t = 0$, the agent $\pi^{(0)}$ is trained to optimize a given task with a text description $l^{(0)}$. Then at any following preference iteration $t \in [1, T]$, a human preference annotator watches a rollout of policy $\pi^{(t-1)}$ and gives language preference $h^{(t)}$. Given the latest policy $\pi^{(t-1)} := \mathcal{A}_M(R^{(t-1)})$ and history of human preference $\{l^{(0)}, h^{(1)}, ..., h^{(t)}\}$, the objective is to output a reward function $R^{(t)}$ such that $F^{(t)}(\mathcal{A}_M(R^{(t)}))$ is maximized. In this problem, $F^{(t)}(\cdot)$ is the ***human's satisfaction level*** with a task execution controlled by $\pi^{(t)} := \mathcal{A}_M(R^{(t)})$. We once again note the importance of maximizing $F^{(t)}$ at every $t$, rather than using multiple $h^{(t)}$ to maximize one $\bar{F}$.

## 3.1 EVALUATION FRAMEWORK

It is ultimately human satisfaction that matters, meaning we need to measure a complex, noisy quantity that itself is given on the policy rollout video, a variable and open-ended stimulus. At the same time, ROSETTA's reward functions need to be evaluated for semantic match and optimizability. We therefore propose three sets of metrics on the set of reward functions detailed as follows.

**Alignment** requires thorough human survey. We propose a) binary and ternary descriptive questions regarding the preference, such as "did the robot incorporate your preference that wasn't met in the previous video," "did the robot contradict any part of your preference" etc. b) Binary and Likert questions that quantify satisfaction (Hyman & Sierra (2016)). Full survey text is in Sec. D.2.1.

**Semantic match** requires establishing that the goal in the generated code matches the goal in the preference. Given that reward code is a long-form generation and therefore difficult to objectively evaluate (Rosati et al. (2024)), we propose expert text evaluation. ROSETTA's preference-code pairs are evaluated by roboticists, also with binary and Likert questions such as "does the reward function miss part or all of the reward plan" or "are there common sense errors". Full survey text is in Sec. D.2.1.

**Optimizability** measures whether the generated reward function is suitable for the learning algorithm $\mathcal{A}_M(\cdot)$. We use the success rate on evaluation rollouts to tell us how well the reward function can be optimized given adequate models and training algorithms.

## 4    APPROACH

We introduce ROSETTA, which is made of three prompt-based modules: Preference Grounding (Sec. 4.1), Staging (Sec. 4.2), and Coding (Sec. 4.3). As seen in Fig. 2, these modules are built on each other to get the final reward function. All prompts are included in Sec. C.

### 4.1    PREFERENCE GROUNDING

Generating rewards from unconstrained human language preference first requires that the preference be contextualized and disambiguated. Foundation models rely on the letter of the prompt, so the system needs to match vague references, consider prior preferences, etc. For example, consider "no, get it to the center." "it" needs to be mapped to `self.obj`. "center" needs to be mapped to "center of `self.target`. Even "get" could be mapped to "push". This preference was given on a reward function that already aimed for the agent to push the ball to the center, so the "no" needs to be recognized as rejection of the performance but not of the goal. Finally, the previous preference was "can it start further back?" and this needs to be retained despite not being mentioned.

We therefore introduce a Preference Grounding module that uses trajectory images, symbolic states, and a single-sentence task description to get per-step language descriptions. From these, it generates a summary of task performance, grounded human preference, and a new single-sentence goal for use in the next interaction step. The above example becomes "the agent did not push the ball to the center of the target as requested. It should do so while still starting further back." Here we use `gpt-4o` (OpenAI et al. (2024a)) given its strong vision-language capabilities.

### 4.2    STAGING

In unconstrained language preference, a single sentence can completely change the task goal without providing guidance on steps to take. However, dense reward often requires breaking the task into stages and rewarding each one. We automate staging by leveraging its similarity to task-planning. foundation models are adept at planning (Huang et al. (2022a)), and unlike many settings that require fixed output structure, here we can allow the foundation model to stage in language space and capture nuances embedded in the language. Given "nestle the ball into the bin", the Staging module generates a plan including "Stage 3: Reward the agent for placing the ball in the corner of the bin with a tight threshold". These function as concrete design notes for the Coding module.

The Staging module takes four inputs: existing environment code as in Ma et al. (2024b), generated task description, generated demo summary, and grounded preference. `gpt-4o` is used here as well.

### 4.3    CODING

The coding module implements the plan created by the Staging module. It takes several key elements: a history containing the staging plan, environment code, summarized documentation on environment attributes, Grounding outputs, desired function headers, and importantly, a checklist of domain knowledge. In ROSETTA, this includes knowledge on densifying reward, masking staged rewards, and considering problem geometry: all crucial to reward success. The full checklist is in Sec. C. For long-horizon action primitive environments, the Coding module generates target positions, dense position reward, and success conditions for each stage. These are inserted into a task-agnostic template to ensure action space adherence. In short-horizon continuous control, all code is generated.

The Coding module uses `o1-mini`. Many foundation models, even vision-language models, hallucinate on spatial reasoning (Li et al. (2024a)). However, reasoning models (OpenAI et al. (2024b)) are known not only for their coding ability but also their ability to apply long contexts and instructions. `o1-mini` achieves this with relatively low cost.

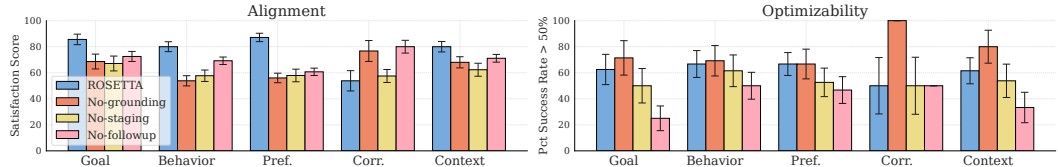

Figure 3: ROSETTA vs baselines, compared on optimizability and alignment for goal, behavior, preferential, corrective, and context-dependent preferences. ROSETTA outperforms on all preferences except corrective, as expected given Eureka and Text2Reward are built for the corrective setting.

Figure 4: ROSETTA vs ablations, compared as in Fig 3. ROSETTA outperforms on alignment metrics for all preferences other than corrective. For optimizability, ROSETTA-no-grounding appears superior because ungrounded preferences often result in minimal edits to the prior reward. If it is optimizable, the success rate will be high despite it being misaligned. See Sec. 5.2.4 for analysis.

**Verification questions for domain knowledge.** While reasoning foundation models are strong, long context can result in information loss (Chuang et al. (2024)). Giving the same information in smaller chunks can promote adherence. At the same time, the foundation model is unlikely to completely change strategies once the initial code has been written. We therefore give domain knowledge in the initial coding prompt, then reiterate in a sequence of verification instructions. We train and evaluate a subset of the resulting variants. As in prior works (Xie et al. (2024b); Pan et al. (2024)), we use **iterative error correction** on the first and final variants. Generated reward code is run in the simulator, with errors corrected by `o1-mini`. Across our main set of 116 preferences, 79.35% of generated reward functions ran before error correction, and 95.38% ran after.

## 4.4 POLICY TRAINING

ROSETTA results in $V * N$ reward functions, where $V$ is the number of variants and $N$ is the number of end-to-end generations. $V$ is fixed based on the number of verification questions, whereas $N$ is a hyperparameter. We train at most three variants per generation given resource constraints. To select $\pi^{(t)}$ at iteration step $t$, we take the best policy from each reward variant's training run and let the preference annotator choose. Consider the set of variants $\mathcal{R}^{(t)} = \text{ROSETTA}(h^{(t)})$. For all $R \in \mathcal{R}^{(t)}$, we take policies $\mathcal{A}_M(R)_{k*}$ where $k^*$ is the training iteration with the highest success rate; if success rate for all training iterations is 0, we use maximum average accumulated reward. The chosen policy is $\pi^{(t)}$. Since the annotator chose it, its reward function $R^*$ is by definition the one that maximizes $F^{(t)}(\mathcal{A}(R)_{k*})$. Given the goal of investigating automated reward generation and selection, we choose hyperparameters *a priori* for each environment, but do not tune for any specific reward function. With sufficient resources, systematic hyperparameter tuning can be integrated.

## 5 EXPERIMENTS

ROSETTA aims to handle natural human preference in a way that is satisfactory to that human. We investigate this through four research questions:

$\mathcal{Q}$1: Does ROSETTA generate adaptive reward functions given a sequence of human preferences?

$\mathcal{Q}$2: Can ROSETTA handle diverse, unconstrained human language?

$\mathcal{Q}$3: Can ROSETTA handle dynamic, unseen preferences, even as they compound over time?

$\mathcal{Q}$4: Does the proposed evaluation framework offer insights beyond standard training metrics?

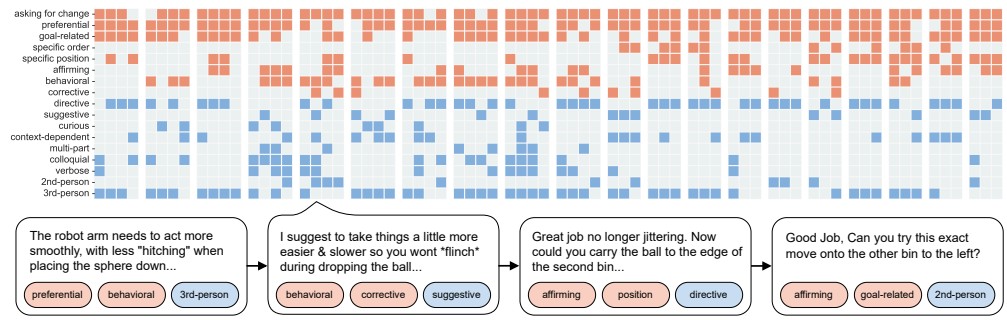

Figure 5: Visualization of diverse human preferences. Each column is one human preference, and each group of columns is a preference sequence. Colored cells indicate applicable categories: orange for language, blue for content. A full list of preference categories can be found in Sec. D.3.

## 5.1 EXPERIMENT SETUP

**Training and environments.** We conduct experiments in two settings. Short-horizon: 7-DoF operational space control, training with proximal policy optimization (Schulman et al. (2017)), two task-agnostic environments (SphereAndBins and BallAndTarget) each with ten preference sequences of length four. Long-horizon: hierarchical policy trained with soft actor-critic (Haarnoja et al. (2018a)) and primitive skills adapted from MAPLE (Nasiriany et al. (2022b)), three task-agnostic environments (ThreeCubes, ObjectsAndBins, ObjectsAndDrawer) each with five preference sequences of length two or three. For both, we use ManiSkill 3 (Tao et al. (2024)) and the Franka Panda (Haddadin et al. (2022)). Annotators are sourced from Upwork (Upwork (n.d.)). Full details in Sec. F.2.

**Baselines.** We compare ROSETTA to Eureka (Ma et al. (2024b)) and Text2Reward (Xie et al. (2024b)), Both use multiple rounds of language feedback to optimize a fixed goal, with the language feedback being purposely written to help shape this reward. Eureka also uses multiple rounds of best-of-n sampling, supervised by hard-coded ground-truth fitness functions. We test each method by inputting a human language preference, then having that human select among policy videos from the generated functions. As noted in Sec. 3, we consider human evaluation to be the fitness function for all methods. We begin with the original ManiSkill reward, then evaluate every method after a single round of application to test immediate adaptation. We compare on both of our short-horizon task-agnostic environments, as well as two more used in Text2Reward that were not seen during ROSETTA's development. We apply the training pipeline described in Sec. 4. `o1-mini` is used for both methods, as they both involve code generation in every generation.

**Ablations.** We compare ROSETTA to three ablations: 1) ROSETTA-no-grounding, in which we give the original human preference directly to Staging. 2) ROSETTA-no-staging, in which we ground the preference and give it directly to a version of Coding with all mentions of "stage" removed. 3) ROSETTA-no-followup, in which we do not apply any follow-up questions.

**Metrics.** For alignment, we consider percent preferences where the annotator was more satisfied and average Likert-scale satisfaction score. For semantic match, we consider seven-point Likert scores for each of Grounding, Staging, and Coding. We multiply these to get a Cascading score. For optimizability, we consider the average training success rate and percent of preferences that have a policy with a success rate $>50\%$. Detailed definitions of each metric are in Sec. E.

## 5.2 RESULTS

### 5.2.1 ROSETTA GENERATES ADAPTIVE REWARDS TO SEQUENCES OF PREFERENCES ($\mathcal{Q}1$)

ROSETTA generates semantically matched, optimizable, and aligned reward functions for sequences of human preferences. As shown in Tab. 2, it improves satisfaction on 86.0% of preferences and achieves a mean satisfaction score of 79.0%. It has even higher success rate and percent success >50%, and similarly successful performance on semantic match; all measured across both short- and long-horizon. Per-interaction step, we see in Tab. 1 that performance is maintained even as context

and changes layer on each other, with at least 80% of preferences resulting in satisfaction and at least 75% satisfaction score in each step. There is some degradation in training metrics and degree of satisfaction, often caused either by the preference annotator doubling down on difficult preferences or hallucinations from prior steps contaminating the current prompt. The latter is a key failure mode of prompting with environment code that includes the prior reward function; examples in Sec. B.3.1.

**Comparing to baselines.** We see in Fig. 3 that ROSETTA outperforms Eureka and Text2Reward, particularly when the preference changes task requirements. We compare on preferences that are *corrective* (give corrections on the same task requirements), *preferential* (add/change task requirements), *goal-related* (address success definition), and *behavioral* (address other parts of reward landscape). While the baselines are similar on corrective preferences, ROSETTA outperforms on task-changing cases: preferential, goal-related, and behavioral. We expect that with the correct (human-defined) fitness function at each step, baselines would improve in performnace, but only if able to generate code that effectively specified the success condition of every preference. Doing so would require ROSETTA's prompt components, making the resulting method a hybrid.

Cost comparison can be found in Sec. G.

To test the value of the approach itself, we test with open-source language models: Qwen2.5-VL-72B-Instruct for grounding/staging and Qwen3-Coder-480B-A35B-Instruct-FP8 for coding. ROSETTA with the proprietary models achieves mean best reward of 0.623; with open-source models, it achieves 0.741, giving preliminary evidence of the value of ROSETTA across LLMs. To test the value of the design choices vs. specific prompts, we test with the prompts paraphrased by `gpt-4o` and Gemini 2.5 (Team (2025)). Gemini 2.5 paraphrasal leads to 0.584 mean best reward, while `gpt-4o` leads to 0.658, suggesting that the specific prompt matters less than the conceptual structure.

### 5.2.2 ROSETTA EXTRACTS SIGNAL FROM DIVERSE, UNCONSTRAINED LANGUAGE ($\mathcal{Q}2$)

Human can communicate preferences with diverse language categories as shown in Fig. 5. ROSETTA has consistent performance across language features that are difficult for foundation models, such as *specific positions* and *context-dependent*, as shown in Figs. 7 and 8. The *specific positions* category has complex and often evolving position descriptions. These are challenging for foundation models given the lack of spatial reasoning (Liao et al. (2024)). The *context-dependent* category includes preferences that require prior context, e.g. a sequence like "red on top" then "other one actually." Without context, the second preference would be impossible to interpret even for a human. ROSETTA generates productive reward functions even in these challenging cases, achieving 60% success rate and 92% satisfaction score on specific position preference, and 73% success rate and 76% satisfaction score on context-dependent preference. Fig. 5 shows the language diversity of collected preferences.

ROSETTA's ability to handle unconstrained language comes from Preference Grounding and Staging, which interpret the language. In Fig. 4, ROSETTA-no-grounding suffers on alignment for context-dependent cases where environment code is insufficient. ROSETTA-no-staging suffers in general.

### 5.2.3 ROSETTA CAN HANDLE DYNAMIC, UNSEEN, COMPOUNDING PREFERENCES.

Besides diverse language features, human preferences also contain diverse content as in Fig. 5. ROSETTA maintains performance across content features based on Fig. 7. Policies reach specific positions ("right edge of left bin") and orientations ("give it a quarter-turn"), and adapt to behavior preferences ("go slower, that's stressful to watch"); more examples in Sec. B.2. We again see ROSETTA's superior ability to handle changing tasks: Tab. 4 shows that ROSETTA outperforms ablations on goal-related, behavioral, and preferential content. ROSETTA-no-followup matches performance only in corrective cases where verifications are distracting. Tab. 1 demonstrates that ROSETTA's performance shows only limited degradation, staying above 75% on all metrics.

Table 1: Breakdown of optimizability and alignment across iterations. Metrics are as described in Sec. 5.1.

| Iter. | Succ. Rate | % Succ. >50 | More Sat. | Sat. Score |
|---|---|---|---|---|
| 1 | 94.6 | 95.2 | 90.0 | 90.0 |
| 2 | 89.4 | 90.0 | 80.0 | 75.6 |
| 3 | 94.4 | 100.0 | 80.0 | 76.9 |
| 4 | 78.1 | 75.0 | 89.5 | 78.9 |

The strength comes from both `o1-mini`'s coding and reasoning capabilities and ROSETTA's domain knowledge. There is no orientation-specific prompting, but `o1-mini` can reason about the quater-

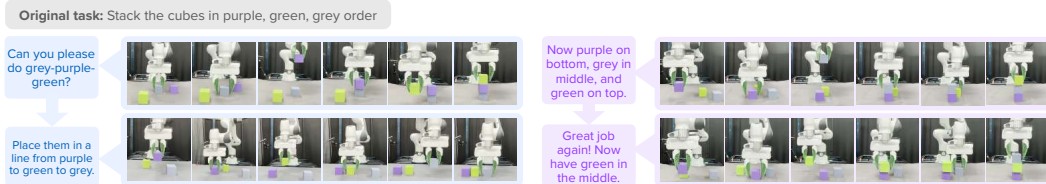

Figure 6: Example real robot results. The first element in each color column represents a preference given to the policy trained on the original task. The second element represents another iteration of preference given on its parent. We see that simulation-trained policies transfer to real effectively.

nions to reward "move the ball in an arc". However, spatial reasoning-heavy preferences such as "corner of the bin" require domain knowledge about how to offset target positions by object dimensions; behavioral preferences require knowledge on how to apply unstaged penalties. Comparison to ROSETTA-no-followup shows that in most cases, review using knowledge is beneficial.

**Fig. 7 demonstrates ROSETTA's content failure modes.** These naturally include preferences that are physically difficult ("throw the ball against the wall and have it bounce back to the bin") or impossible ("pick up the ball instead of pushing" when the ball is too big for the gripper). It also includes preferences that are impossible to represent in ManiSkill's default reward formulation, which does not include temporal info, such as *release*: preference annotators often ask explicitly for a ball to be dropped into a bin from a height. Without any temporal info, this requires an "ungrasp" reward that has no option but to be conditioned on the ball being above the bin. The moment the gripper ungrasps the ball, it stops being "above" the bin and the agent loses reward. See Sec. B.3.

### 5.2.4 EFFECTIVENESS OF THE EVALUATION FRAMEWORK ($\mathcal{Q}4$)

The proposed evaluation framework investigates performance more deeply than standard metrics alone. ROSETTA is performant on all metrics as shown in Tab. 2, but overlap is not a given: 73% of human-selected policies have $\geq 50\%$ success rate, and only 66% have the highest success rate across all options. Optimizability metrics retain practical value in assessing whether ROSETTA is technically reliable: high success rates indicate behavior that is expected from the reward function.

**Low-alignment high-optimizability** rewards appear when the reward code effectively induces its own success condition, but the success condition itself does not match the preference: a semantic hallucination. A common example: ROSETTA is given a preference with nonstandard phrasing and mostly ignores it, outputting a barely-edited version of the previous reward function. The result is a good success rate but a behavior that does not reflect the input. This is part of the reason the ROSETTA-no-grounding ablation, in which Staging and Coding receive more nonstandard phrasing, has high optimizability but significantly lower alignment than ROSETTA. It does match ROSETTA's alignment on corrective preferences, where not changing the success condition is ideal.

**High-alignment low-optimizability** rewards demonstrate deeper challenges. We find that without alignment, we wouldn't know the annotator's true opinion; but without optimizability we don't know the reasons for failure. For example, given preference to "push the ball on the table next to the bin", the human-selected rollout shows a gentle, ineffectual push toward the bin with a success rate of 0; the high-success rate policies pick and place instead of pushing. **Key misalignments:** for the annotator, the push is more important than the position, but this is unclear from the language. Second, ROSETTA set a specific target point because this is typically more optimizable than a target region, but here that point was difficult to push to. Third, the reward function *did* encourage a closed gripper, but the success boost outweighed the opening penalty.

### 5.2.5 REAL ROBOT EXPERIMENTS

We take a sim-to-real approach for real robot experiments. We test on four two-iteration preference chains in the ThreeCubes long-horizon task-agnostic environment for which policies have been trained in simulation. We track positions of the cubes from a single RGB image using FoundationPose (Wen et al. (2024)), then populate the state dictionary required for our long-horizon action primitive setting. This results in an observation equivalent to ManiSkill's standard `state_dict`. The average first

iteration success rate is 72.5%; the average second iteration success rate is 80.0%. Examples are shown in Fig. 6. We observe no sim-to-real gap from the policy.

## 6 LIMITATIONS AND BROADER IMPACTS

One key limitation of our work is transfer to other simulators. ROSETTA is task-agnostic, but it would help to know the effort required to transfer domains. Another limitation is the expert evaluation for Semantic Match metrics. Analytic/learned metrics may be cheaper and more objective.

This paper enables adaptation of RL-based embodied agents to human language preferences, and uses foundation models to do so. Certain human preferences may be less represented in a foundation model's domain and result in less aligned or performant rewards. Furthermore, preventing a harmful generation is an unresolved ethical and technical question. This paper is designed to be an asset that empowers users: frameworks like this one can and should be fitted with further domain-specific safety measures as they advance. We design ROSETTA to be easily extendable to more domain knowledge.

## 7 CONCLUSION

We aimed to investigate how reward signals can be extracted from unconstrained, diverse, and dynamic human language to train embodied AI agents. To this end, we present ROSETTA. ROSETTA takes in such language, extracts desired task content, and generates useful reward functions even when that content changes and contradicts prior content. Given the inherent challenge of evaluation under unknown, subjective ground truth, we present an interactive task structure and metrics of both alignment and objective efficacy. ROSETTA is demonstrated to be semantically matched with preference, optimizable, and aligned with human preference.

ACKNOWLEDGMENTS

This work is in part supported by the Stanford Institute for Human-Centered AI (HAI), AFOSR YIP FA9550-23-1-0127, ONR N00014-23-1-2355, ONR MURI N00014-22-1-2740, and ONR MURI N00014-24-1-2748.

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

| | # Preferences | Semantic Match | | | | Optimizability | | Alignment | |
|---|---|---|---|---|---|---|---|---|---|
| | | Grounding | Staging | Coding | Cascading* | Success Rate | Pct Success > 50 (%) | Satisfied (%) | Satisfaction Score |
| Short Horizon | 80 | 95.7 | 87.3 | 84.6 | 78.1 | 89.2 | 90.1 | 84.8 | 76.4 |
| Long Horizon | 36 | 80.0 | 90.0 | 82.9 | 78.6 | 82.2 | 88.8 | 88.8 | 92.4 |

Table 2: Experimental results evaluating semantic alignment, optimization potential, and user satisfaction. Semantic match metrics: expert Likert evaluation on grounding quality, staging quality, coding quality, cascading (product of all three). Optimizability metrics: Success rate: average success rate of best policy for a single preference, Pct Success >50%: percent runs with success rate above 50%. Alignment: Satisfaction: % preferences where annotator was satisfied, Satisfaction Score: preference annotator Likert evaluation on how much more/less satisfied (50 is neutral). **Note:** Cascading Match excludes cases where expert evaluators found the feedback difficult to interpret.

## A ADDITIONAL RESULTS

We see in Fig. 7 and Fig. 8 that ROSETTA performs consistently on various types of language and content, as referenced in the main text. This is maintained in challenging cases like context-dependent language and behavioral feedbacks where reward strategies are nonobvious. It also demonstrates that alignment and optimizability are not the same, but do relate.

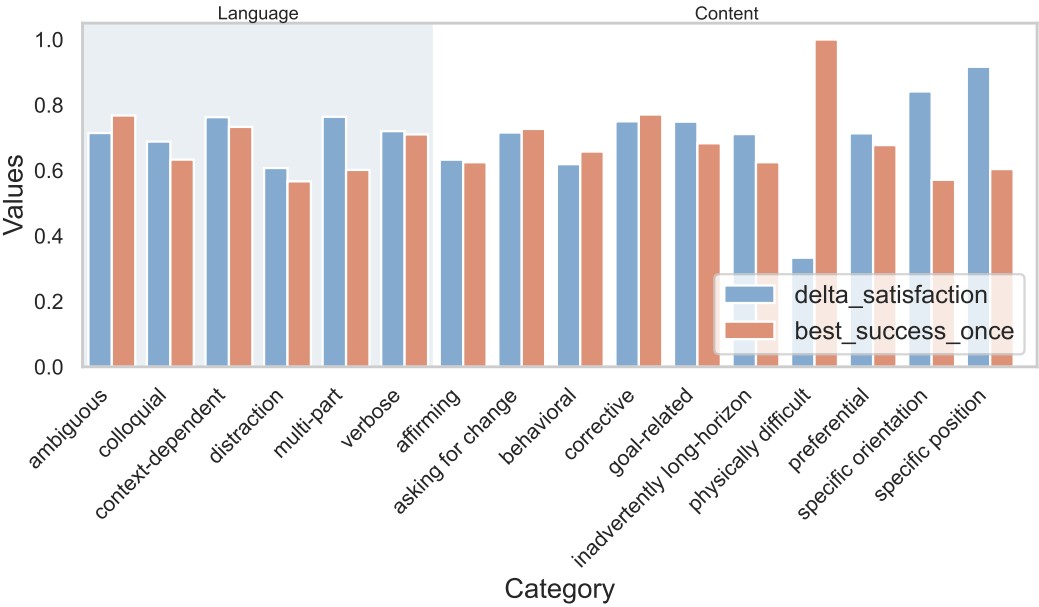

Figure 7: ROSETTA performance on various categories on **short-horizon** environments. For legibility reasons, we exclude some language categories that foundation models are able to handle without additional domain knowledge, such as 3rd-person POV, 2nd-person POV, no POV, curious tone, directive tone, and suggestive tone. We also exclude conditional and physically impossible feedbacks, which are rare.

## B EXAMPLES

In this section, we present examples illustrating how our pipeline processes human feedback, along with its limitations. Additionally, we provide examples demonstrating the necessity of a multi-faceted evaluation framework.

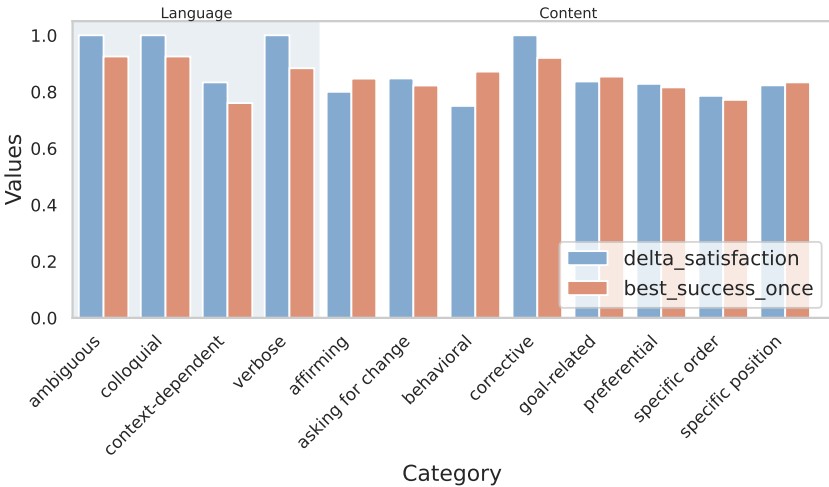

Figure 8: ROSETTA performance on various categories on **long-horizon** environments. For legibility reasons, we exclude some language categories that foundation models are able to handle without additional domain knowledge, such as 3rd-person POV, 2nd-person POV, no POV, curious tone, directive tone, and suggestive tone. We also exclude conditional, inadvertently long-horizon, physically difficult, distraction, multi-part, and physically impossible feedbacks, which didn't occur in long-horizon experiments.

### B.1 SUCCESSFUL CASES IN SHORT-HORIZON

Here, we present a sequence of interactions between our pipeline and the annotator, where ROSETTA successfully adapts the policy to the human's feedback and makes continuous progress.

- **Initial Demonstration:** Robot pushes the ball to the goal.

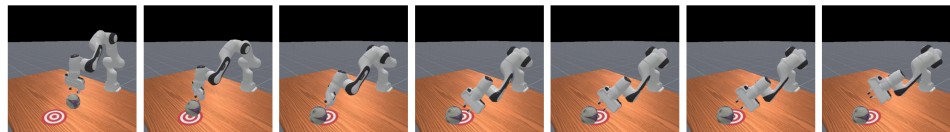

- **Step 1 Human Feedback:** I like the way the robot uses its claw to push force on the sphere to move it. I want to the robot to push the sphere rightward on the table, but as far as the robot is able to.
- **Step 1 Generated Grounded Feedback:** Push the ball as far to the right on the table as possible.
- **Step 1 Generated Goal:** Push the ball as far to the right on the table as possible.
- **Step 1 Demonstration:** The robot pushes the ball all the way to the right

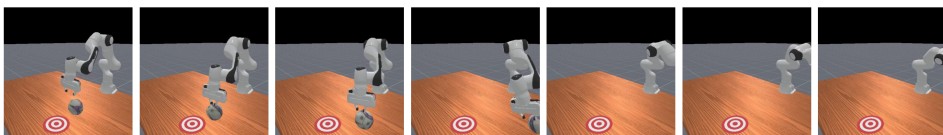

- **Satisfaction Score:** 87.5

- **Step 2 Human Feedback:** I can see the "effort" of the robot trying to push the ball right. I want to see the robot push the ball in the best spinning motion it can do.
- **Step 2 Generated Grounded Feedback:** Push the ball using a spinning motion.

- **Step 2 Generated Goal:** Push a ball to goal position using a spinning motion.
- **Step 2 Demonstration:** The robot spins the ball sideways, pushes the ball to the right.

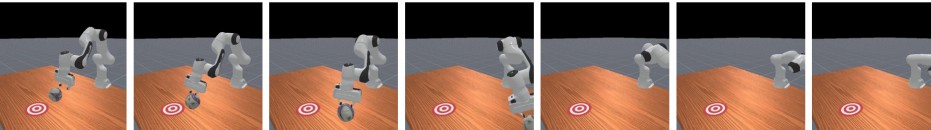

- **Satisfaction Score:** 75.0

- **Step 3 Human Feedback:** I like the effort given to perform the task. Although it was not complete, I enjoyed it. It was a hard task. It kind of rolled the ball in a big circle - maybe it could try using its gripper to just rotate the ball?
- **Step 3 Generated Grounded Feedback:** Use the gripper to rotate the ball directly towards the goal instead of moving it in a circular path.
- **Step 3 Generated Goal:** Push a ball to goal position by rotating it directly towards the goal using the gripper.
- **Step 3 Demonstration:** The robot pushes the ball towards the right of the target while rotating it sideways.

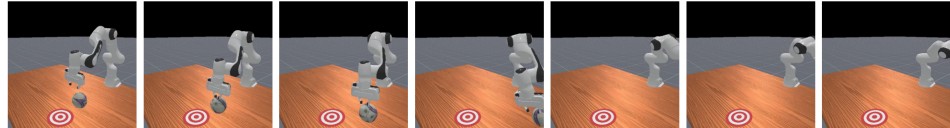

- **Satisfaction Score:** 50

- **Step 4 Human Feedback:** I do not want the robot to try to push the ball to the goal position anymore. The robot performed good prior. I want it to try to make the ball rotate in place, maybe by twisting its gripper on top. I want to see the ball spin.
- **Step 4 Generated Grounded Feedback:** Stop trying to push the ball to the goal position. Instead, make the ball rotate in place by twisting the gripper on top of it to see the ball spin.
- **Step 4 Generated Goal:** Make the ball rotate in place by twisting the gripper on top of it to see the ball spin.
- **Step 4 Demonstration:** The robot slowly rotates the ball by twisting the gripper.

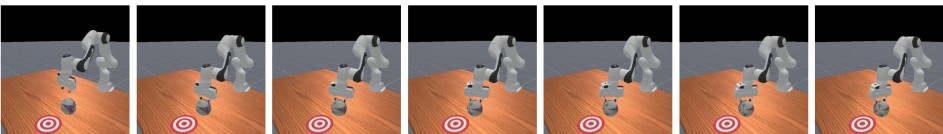

- **Satisfaction Score:** 100

## B.2 SUCCESSFUL CASES IN LONG-HORIZON

Here, we showcase instances where our pipeline successfully handles feedbacks to long-horizon environments.

**Example 1: Success Rate: 0.9, Satisfaction Score: 100**

- **Human Feedback:** Place the light colored items in one bin and the dark objects in the other.
- **Feedback Categories:** no pov, context-dependent, preferential, goal-related , specific position, directive, asking for change.

- **Generated Grounded Feedback:** Place the apple and orange in one bin, and the baseball and tennis ball in the other bin.
- **Previous Goal:** Place the apple, orange, baseball, and tennis ball all into the same bin.
- **Generated New Goal:** Place the apple and orange in one bin, and the baseball and tennis ball in the other bin.
- **Previous Demonstration:** The robot picks up the objects and puts them into the white bin one by one.

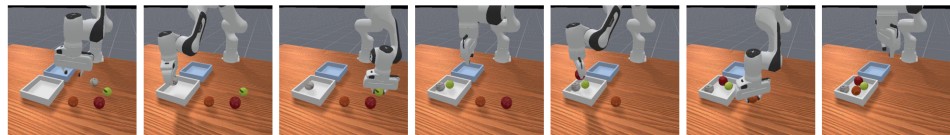

- **New Demonstration:** The robot placed the baseball and tennis ball in the white bin, and apple and orange in the blue bin.

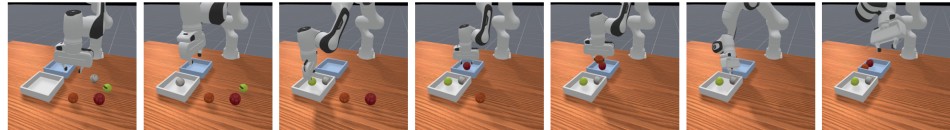

### B.2.1 EXAMPLE 2: SUCCESS RATE: 1.0, SATISFACTION SCORE: 100

- **Human Feedback:** Can the robot pick up the other items on the floor? It doesn't appear complete with its task leaving the other items there.
- **Feedback Categories:** 3rd-person, verbose, suggestive, preferential, context-dependent, corrective, asking for change.
- **Generated Grounded Feedback:** Pick up the red soup can and place it in the drawer.
- **Previous Goal:** Put apple in drawer.
- **Generated New Goal:** Put Apple and Red Soup Can In Drawer
- **Previous Demonstration** The robot picks up the apple and places it in the drawer.

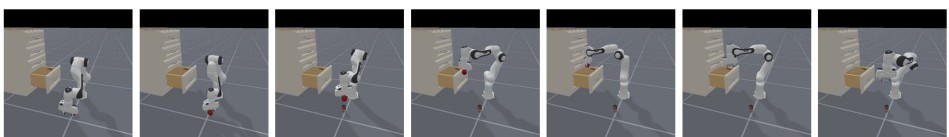

- **New Demonstration** The robot placed the apple in the drawer, then the soup in the drawer.

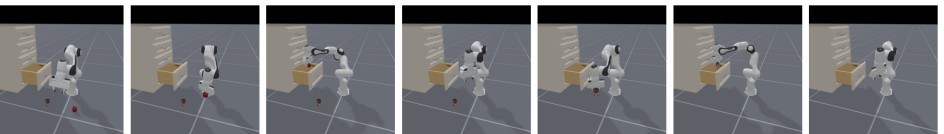

### B.2.2 EXAMPLE 3: SUCCESS RATE: 0.6, SATISFACTION: 100

- **Human Feedback:** I am glad the robot was able to completely stack the three cubes in the correct order. I would like the robot to now push the green cube next to the pink cube. After, I want the robot to pick up the red cube and drop it on top of the two cubes.
- **Feedback Categories:** 3rd-person, preferential, behavioral, specific order, specific position, goal-related, asking for change.

- **Generated Grounded Feedback:** Move the green cube next to the purple cube. Then, pick up the red cube and place it on top of the green and purple cubes.

- **Previous Goal:** Stack the green cube on top of the red cube, then stack the purple cube on top of the green cube.

- **Generated New Goal:** Move the green cube next to the purple cube. Then, pick up the red cube and place it on top of the green and purple cubes.

- **Previous Demonstration:** The robot stacks three cubes together.

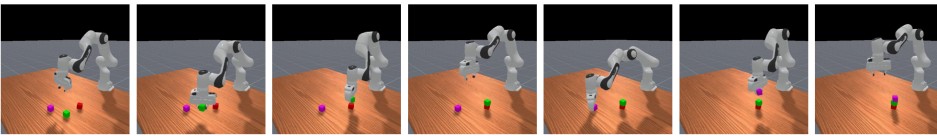

- **New Demonstration:** The robot pushes the green cube next to the purple cube and places the red cube on top of them.

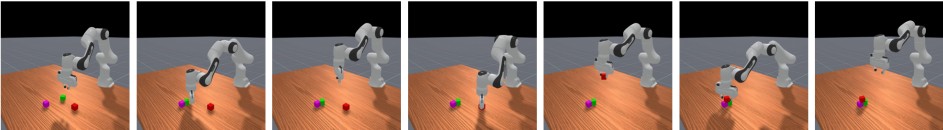

## B.3 EXAMPLES OF LIMITATIONS

### B.3.1 LIMITATION 1: MISTAKE ACCUMULATION

A specific failure case we observed occurs when the language model propagates errors from the current reward function to the new one. In this case, the LLM mistakenly added a +0.1 offset to the z-coordinate of the target position. In the previous step, due to a correctly defined success condition and some luck, the robot was able to successfully learn the task. However, this error was inherited in the generated reward function, leading to unsuccessful learning in the subsequent step.

- **Previous Reward Function Snippet:**

```
def stage_0_reward():
    # Target position computation
    # Action: pick up the red apple
    # Target position is slightly above the apple to approach from above
    apple_pos = prev_info['apple_pos']  # [x, y, z]
    target_pos_1 = apple_pos.copy()
    target_pos_1[2] += 0.1  # Raise z by 0.1 meters above the apple
```

- **Generated Next Step Reward Function Snippet:**

```
def stage_0_reward():
    # Target position computation
    # Action: pick up the red apple
    # Target position is slightly above the apple to approach from above
    apple_pos = prev_info['apple_pos']  # [x, y, z]
    target_pos_1 = np.array([
        apple_pos[0],
        apple_pos[1],
        apple_pos[2] + 0.1  # 0.1 meters above the apple
    ])
```

- **Previous Demonstration:** The robot puts the two objects in the drawer and closes it.

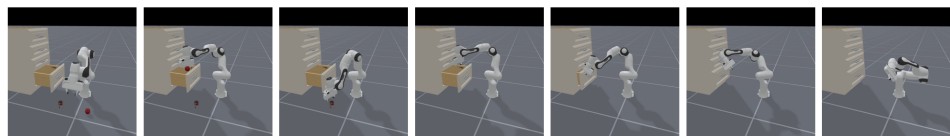

- **New Demonstration:** The robot repeatedly tries to pick up the apple but doesn't lower the gripper enough.

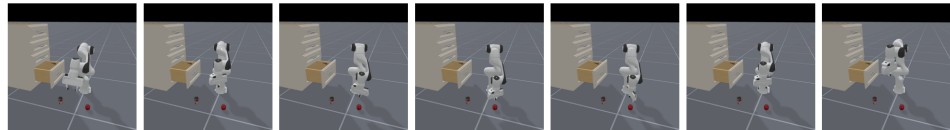

The flawed reward function ended with a 0 success rate and a 0 satisfaction score.

### B.3.2   LIMITATION 2: CONTENT CONSTRAINT

Inadvertent long-horizon scenario.

- **Step 1 Feedback:** I want the robot to release the sphere after placing it.
- **Step 1 Demonstration:** The robot lifts the cube above the bin but fails to release it.

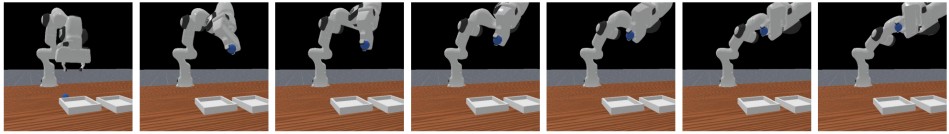

- **Step 2 Feedback:** I want the robot to put the ball in the corner of the bin
- **Step 2 Demonstration:** Still, the robot lifts the cube above the bin but fails to release it.

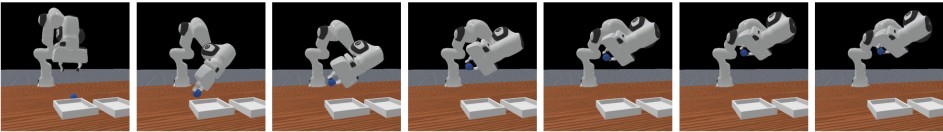

- **Step 3 Feedback:** I want it to not try to first bring the ball above the corner then release. It seems that's not going to work. It should put it straight down into the bin corner.
- **Step 3 Demonstration:** The robot picks up the ball and moves it to the corner of the bin. Though this is listed as an example of a fundamental limitation, this final adaptation despite contaminated history shows the power of the modular framework to recover and adapt to helpful human input.

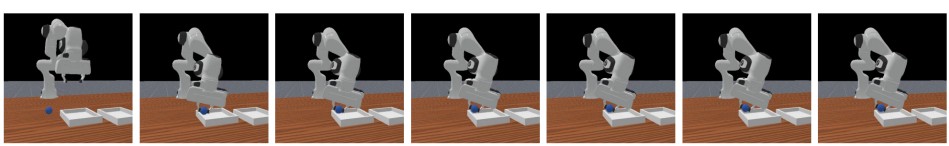

## B.4 Constraints

While ROSETTA accepts unconstrained language, we established certain content constraints for annotators, though we accepted some violations to show the system's limitations. Constraints:

- No temporal dependency. Since the state history is not given to the reward function except one step back in the action primitive case, we disallowed preferences referencing history. An example: "pause your gripper 2cm above the sphere, then continue down." This cannot be implemented with history-agnostic masking. The most successful agents paused 2cm above the sphere and never progressed.

- No long-horizon tasks in the continuous control setting - described above. This was violated many times as it can be nonobivous to an annotator.

- To some degree, please accept jitter. Stabilization is possible, but it's not the default behavior. If you want stabilization, please say so explicitly.

- Avoid asking for multiple goals at once (multi-objective reward). This was rare regardless, as annotators quickly recognized that asking for two things at once was unrealistic.

- Avoid asking for ungrasp in continuous control: this turns out to be difficult to learn since grasping is usually the right move. The algorithm did learn it in many cases, but this was more due to its own sampling leading to a success boost than the dense ungrasping reward.

- Action primitive setting only: can only give feedback on positions, cannot ask for pulling action.

## C Prompts

### C.1 Short Horizon Reward Generation Prompts

To generate the reward function for a short-horizon environment, we follow the steps outlined below. The entire pipeline takes the following inputs:

- **Current Environment Code:** The full ManiSkill environment code, including the current reward function.
- **Grounded Feedback:** Grounded human feedback.
- **Demo Summary:** A generated language description of the demonstration.
- **New Task Goal:** The updated task goal based on the feedback.

The generation process is carried out through a multi-round conversation with GPT-4o, following these steps:

1. **Staging:** In this step, the LLM is tasked with planning the reward function based on the feedback. (Prompt 1)
2. **Coding:** The LLM is tasked with generating reward function code snippets according to its plan. (Prompt 2)
3. **Error Correction 1:** We execute the generated code and ask LLM to regenerate the reward function with the error trace if the code fails.
4. **Geometry Review:** The LLM reviews the generated code and corrects any geometry-related mistakes. (Prompt 3)
5. **Target Position Reward Review:** The LLM reviews and corrects target position setting mistakes in the generated code. (Prompt 4)
6. **Reward Design Review:** The LLM reviews and corrects reward function design mistakes in the generated code. (Prompt 5 and (Prompt 6)
7. **Error Correction 2:** We execute the generated code and ask LLM to regenerate the reward function with the error trace if the code fails.

**Prompt used:**

### Short Horizon Staging Prompt

```
# Instructions

You are a reward engineer that is an expert at designing reward
functions to solve reinforcement learning tasks. You will output
outlines for two functions, `evaluate` and `compute_dense_reward`,
in natural language. They should be concrete and ready for
implementation.

You will be given:
1. Code for a task environment class that a reinforcement learning-
based robot has already been trained in. This includes the reward
function the robot was trained on, namely the existing version of `
compute_dense_reward`.
2.  Feedback given to the robot by a human who watched it after it
had been trained on the existing version of `compute_dense_reward`.

You should:
1. Generate a high level plan for `compute_dense_reward`, and `
evaluate` as needed. `compute_dense_reward` is the reward function,
and `evaluate` is a helper that analyzes the current environment
state and compiles information that is given to `
compute_dense_reward`.
2. Design a staged reward:
  - Split the task into stages and give the agent reward gradually,
 encouraging it to complete each stage. The reward should therefore
 accumulate, NOT be all-or-nothing at the end.
  - Consider interdependencies and tradeoffs between different task
 stages and different aspects of the feedback and overall goal.
Ensure your reward design isn't counterproductive.
3. If there are aspects of the feedback that it's impossible to
incorporate without modifying other methods, say "I cannot do <
aspect>" and ONLY incorporate the other parts of the feedback, if
there are any.
4. If there are aspects of the feedback that are physically
impossible, say "I cannot do <aspect>" and ONLY incorporate the
other parts of the feedback, if there are any.
5. Explain your reasoning at each step.

Each stage must be a **single outcome**. It must be a change in an
environment state, not a particular action. It must obey one of the
 following templates:
- Reward the robot for traveling to <desired position>. Example: "
reward the robot for traveling to the point just left of `cubeA`."
- Reward the robot for getting <desired object> to <desired
position>. Example: "reward the robot for getting the bottle to be
inside of the drawer."
- Reward the robot for <other single outcome>. Example: "reward the
 robot for moving more smoothly."
- Reward the robot for <other single descriptor> <other object>.
Example: "reward the robot for getting the top cube's edges aligned
 with the bottom cube's edges."
Tips:
- Notice how the stage templates are highly atomic - one single
outcome.
- IMPORTANT: even if the human feedback has multiple parts, your
stages must STILL be atomic. Just because the person lists multiple
 parts, this doesn't mean each part is a single stage. One part may
 still require multiple atomic stages, and one atomic stage may
contribute to multiple feedback parts. BE SMART, DON'T JUST LIST
OUT THE FEEDBACK AS YOUR STAGES.
```

```
- Note how the stage templates deal with changes to the environment
  (including the robot) rather than the robot's actions.

### Details for `evaluate`
```python
def evaluate(self: BaseEnv) -> Dict[str, torch.Tensor]
    """
    Return dict is the dict mapping strings of reward-relevant
questions to their boolean-valued answers for the batch. So, the
values of this dict are torch.Tensors with bool dtype, where the
first dimension is a batch of episodes and the second is the
boolean answer to the string question for that individual episode.

    Should create a useful set of information. `evaluate` will be
called on both the previous state (before the agent took an action)
 and the current state (after the agent took an action) to
calculate reward.
    """
```

### Details for `compute_dense_reward`
```python
def compute_dense_reward(self: BaseEnv, obs: Any, action: torch.
Tensor, info: Dict[str, torch.Tensor]) -> torch.Tensor
    """
    Encodes reward for each possible action based on `evaluate`
output dictionary and other current environment info (environment
instance attributes).

    Incorporates human feedback as given.

    Obeys the following structure:
    1. Stage reward
       - Task is split into stages and reward is given to the agent
gradually, encouraging it to complete each stage. The reward
accumulates.
       - Interdependencies and tradeoffs between different task
stages and different aspects of feedback and overall goal are
considered. Reward design is not overall counterproductive to meet
short-term goals.
    2. Extra success bump for successful episodes
    3. Return reward value
    """
```

Original code:
```python
{ENVIRONMENT CODE}
```

Current task description: {NEW TASK GOAL}
Robot execution description: {DEMO SUMMARY}
Human's feedback: {GOUNDED FEEDBACK}

## Instructions
Think step-by-step to make a high-level plan for rewriting `
evaluate` and `compute_dense_reward`. The plan should be a series of
 stages.
- Don't code yet, just plan the stages.
- Consider dependencies and conflicts between task stages, and
create a plan that doesn't do anything counterproductive.
- Carefully consider the physical aspects of the task.
  - Plan realistic paths
```

```
  - Consider the physical state the robot is in after each stage.
Write the next stage to build directly on it.

## RULES YOU MUST FOLLOW:
1. Incorporate human feedback as given.
2. Do not add anything that contradicts the feedback.
3. Do not invent your own changes or speculate about what the
feedback is going for.
4. Do not add anything counterproductive to the overall goal.
5. Make every stage atomic.
6. Make the plan complete - think about all the atomic stages the
robot needs to achieve every part of the feedback, not just the
final ones.
7. NEVER ask the robot to release an object from the gripper.
8. NEVER ask the robot to first move above the desired location,
below desired location, then release or lower down or move up. Just
 have it go straight to the desired location.
```

## Short Horizon Coding Prompt

```
# Instructions:
1. Code the reward function based on the stages.
  a. Use the Code Block below wherever possible.
  b. Follow the Checklist below.
2. Only output methods you are editing.
3. If you are editing a method, output the whole edit method, not
just your edits.
4. Don't introduce new methods, not even helper methods. Just edit
`compute_dense_reward` and `evaluate`.
5. Comment your code as needed.
6. If there are aspects of the feedback that it's impossible to
incorporate without modifying `_load_scene` or `_initialize_episode
`, say "I cannot do <aspect>" and ONLY incorporate the other parts
of the feedback, if there are any.
7. Explain your implementation.
8. Explain checklist adherence. Be concise.

1. Code the reward function on the stages. Write a short chain of
reasoning before your code, to explain your reasoning.
  a. Use the Code Block below wherever possible.
  b. Follow the checklist below.
  c. Comment your code as needed.
2. Only output methods you are editing.
3. If you are editing a method, output the whole edit method, not
just your edits.
4. Don't introduce new methods, not even helper methods. Just edit
`compute_dense_reward` and `evaluate`.
5. If there are aspects of the feedback that it's impossible to
incorporate without modifying `_load_scene` or `_initialize_episode
`, say "I cannot do <aspect>" and ONLY incorporate the other parts
of the feedback, if there are any.

# Code Blocks for `compute_dense_reward`

### Make Robot Reach Target Positions
In `compute_dense_reward`:
- **Minimize the distance** between the object's current position
and the target position in the reward function.
```

- **Normalize and smooth distances** using functions like `torch.tanh`:
  ```python
  transport_reward = 1.0 - torch.tanh(torch.linalg.norm(target_pos - obj.pos()))    # max 1.0, decreases with distance
  ```
- **Don't just give a boost when the target position has already been reached.** The robot will get no guidance during movement and be unable to begin.

# Function signatures

## Details for `evaluate`:
```python
def evaluate(self: BaseEnv) -> Dict[str, torch.Tensor]
    """
    Return dict is the dict mapping strings of reward-relevant questions to their boolean-valued answers for the batch. So, the values of this dict are torch.Tensors with bool dtype, where the first dimension is a batch of episodes and the second is the boolean answer to the string question for that individual episode.

    Should create a useful set of information. `evaluate` will be called on both the previous state (before the agent took an action) and the current state (after the agent took an action) to calculate reward.

    Always contains a key called "success" that maps to the success condition
    """
```

## Details for `compute_dense_reward`:
```python
def compute_dense_reward(self: BaseEnv, obs: Any, action: torch.Tensor, info: Dict[str, torch.Tensor]) -> torch.Tensor
    """
    Encodes reward for each possible action based on `evaluate` output dictionary and other current environment info (environment instance attributes).

    Incorporates human feedback as given.

    Obeys the following structure:
    1. Stage reward
       - Task is split into stages and reward is given to the agent gradually, encouraging it to complete each stage. The reward accumulates. Each stage's reward is dense wherever possible.
       - Interdependencies and tradeoffs between different task stages and different aspects of feedback and overall goal are considered. Reward design is not overall counterproductive to meet short-term goals.
    2. Extra success bump for successful episodes
    3. Return reward value
    """
```

# Reward Function Checklist

### Coding Best Practices

- **Use `.clone()`** when calculating and mutating `torch.Tensor`s to avoid unintended reward changes

```
### Selecting Target Positions

- When defining target positions, **specify all coordinates (x, y,
z)**. Do not leave any unrestricted.
- `<element>.pos()` returns the **center position** of the element.
Adjust with offsets if targeting the top, bottom, or sides.
- **Understand geometry and dimensions** by reading the environment
 code.
  - **Example:** If placing an object inside a box, account for
wall thickness.
  - **Example:** If aligning an object with the edge of a target,
consider the target's shape.

### Staged Reward Masking

- **Use torch masking** to activate rewards for a stage **only
after** its prerequisites are met.
- **Example**: Two stagesâĂŤ(1) grasp `bottleA`, (2) move `bottleA`
to `boxA`
  ```python
  # Assume 'is_bottleA_on_boxA' and 'is_bottleA_grasped' are in
info and the target position for `bottleA` is defined
  # Stage 2 reward - moving `bottleA` to `boxA`
  move_reward = torch.tanh(torch.linalg.norm(target_pos - bottleA.
pos()))

  # Reward moving `bottleA` to its target position `target_pos` only
 after it's grasped
  reward[info['is_bottleA_grasped']] += move_reward[info['
is_bottleA_grasped']]
  ```
- **Mask based on completed stages**, not the current one. Example
of bad masking:
  ```python
  # Incorrect: Masking on the current stage's completion
  reward[info['is_bottleA_on_boxA']] += move_reward[info['
is_bottleA_on_boxA']]
  # This fails because it doesn't reward reducing the distance
between `bottleA` to `boxA` until the distance is zero. The robot
therefore doesn't get dense process reward.
  ```
- **Don't reward conflicting stages simultaneously**.
- **Maintain non-decreasing rewards** to encourage progression.
  - **Example:** If the robot must grasp `objectA`, place it,
ungrasp, then grasp `objectB`:
    - Continue rewarding the grasp of `objectA` even when it's being
 placed.
    - Mask the reward for grasping `objectB` based on `objectA`
being at the target location, not whether `objectA` is grasped.

### Reward Component Weighting

- **Equalize the maximum values and shapes of each stage's reward
components.**
- **Prevent reward imbalance for different stages** or the robot
may fail to progress on the task because the additional reward is
relatively small.

### Success Boost
```

```
- **Add a reward boost** when the task is successfully completed to
 emphasize successful episodes.
  - Ensure `info['success']` is a boolean indicating success.
  - Example code:
    ```python
    success_boost_val = 5.0  # Adjust to be 1/4 of total possible
reward so far
    reward[info['success']] += success_boost_val
    ```

{MANISKILL ENVIRONMENT DOCUMENTATION}
```

Short Horizon Geometry Review Prompt

```
- Verify that for each object, this code handles the fact that `.
pos()` returns the center location. Common pitfalls:
  - Not applying the right offsets when placing two objects
relative to each other. E.g. half-width, radius, etc.
- Verify that this code considers the physical attributes of each
object and how they might offset target positions. Common pitfalls:
  - Thickness of box walls
- Verify that this code uses x as the front-back axis and y as the
left-right axis. z is still the up-down axis.
- Verify that when applying penalties, this code does not OVER-
penalize. For example, if slowness is required, set a low upper
bound for speed, but do not penalize it entirely or the robot will
be stalled.

1. Write out your verification step-by-step.
2. Edit the code as needed according to your explanation. Comment
your changes.
3. Only output methods you are editing.
4. If you are editing a method, output the whole method, not just
your edits.
5. Don't introduce new methods, not even helper methods. Just edit
`compute_dense_reward` and `evaluate`.
```

Short Horizon Target Position Review Prompt

```
- Verify that each of the three coordinates is considered for every
 position and offset in the code. Common pitfalls:
  - Forgetting the z-component
  - Ignoring one or two coordinates when setting a target position.
 If the robot is not given constraints in some axis, it could do
anything, leading to a crash into something else.
- Verify that specific target positions are always set. Common
pitfalls:
  - Simply incentivizing the robot to go in a certain direction (e.
g. "down" or "below an object"). It should always be pushed toward
a SPECIFIC POINT IN SPACE, or it won't stop and will fly wildly.

1. Write out your verification step-by-step.
2. Edit the code as needed according to your explanation. Comment
your changes.
3. Only output methods you are editing.
4. If you are editing a method, output the whole method, not just
your edits.
5. Don't introduce new methods, not even helper methods. Just edit
`compute_dense_reward` and `evaluate`.
```

Short Horizon Reward Design Review Prompt

```
- Verify that the reward is *dense* when, and ONLY when, you want
the robot to gradually approach a certain state. "Dense" means a
continuous function that gives more and more reward as the robot
approaches the right position, rather than a step function only
reward it once it's reached.
  - Distance, angular difference, velocity - all continuous values
that can have dense, continuous rewards
  - Traveling to a location gradually and opening/closing a gripper
 to a certain point gradually - ALWAYS DENSE.
- Verify that all *penalties* are NOT dense. Examples:
  - To slow the robot down, there should be a penalty on speed that
 is NOT dense. You want it to just stay below a specific speed, not
 get more reward the slower it is.
  - TO HAVE THE ROBOT GO SLOW, THE CODE SHOULD SIMPLY SET A
CONSTANT UPPER BOUND ON SPEED. DENSE REWARD DOES NOT HELP HERE.
  - To have the robot to keep its gripper closed, the penalty on
gripper opening should not be dense - you want it to stay below a
certain opening, not get more reward the more closed it is.
- Verify that the code actually requires the robot to reach its
target position. Common pitfalls:
  - Defining a "near" threshold that is larger than the "at"
threshold for a target position, then not requiring the agent to
move once it's "near" even if it's not "at".

1. Write out your verification step-by-step.
2. Edit the code as needed according to your explanation. Comment
your changes.
3. Only output methods you are editing.
4. If you are editing a method, output the whole method, not just
your edits.
5. Don't introduce new methods, not even helper methods. Just edit
`compute_dense_reward` and `evaluate`.
```

Short Horizon Masking Review Prompt

```
- Verify that the reward **never decreases** during the progression
 of the task, so that the robot isn't disincentivized from making
progress.
  - The code should only mask on genuine prerequisites
  - The reward should be enabled even when the stage is complete
  - For example, if the robot needs to grasp an object to carry it
to a location and then let it go, give the grasp reward not only
when the gripper is ready to grasp, but also after the object has
been carried to the location and ungrasped.
- Verify that no stage reward components are masked prematurely.
  - Example: reward component guides the robot to put `objA` next to
 `objB`. Masking this with `info["is_objA_next_to_objB"]` means it'
ll only get reward after it succeeds, so it will never get started.
  - Example: reward component guides the robot to travel to point `
target_pos`. Masking this with `info["is_near_target_pos"]` means it
'll only get reward after it succeeds, so it will never get started.

1. Write out your verification step-by-step.
2. Edit the code as needed according to your explanation. Comment
your changes.
3. Only output methods you are editing.
4. If you are editing a method, output the whole method, not just
your edits.
```

```
5. Don't introduce new methods, not even helper methods. Just edit
`compute_dense_reward` and `evaluate`.
```

## C.2 LONG HORIZON REWARD GENERATION PROMPTS

To generate the reward function for a long-horizon environment, we follow the steps outlined below. The entire pipeline takes the following inputs:

- **Current Environment Code:** The full ManiSkill environment code, including the current reward function.
- **Grounded Preference:** Grounded human preference.
- **Demo Summary:** A generated language description of the demonstration.
- **Current Task Goal:** The task goal that the policy is currently attempting to achieve.
- **New Task Goal:** The updated task goal based on the preference.
- **Environment Information Key:** Dictionary Keys of the observation space, with corresponding descriptions.
- **Environment Setup Description:** A description of the environment setup.
- **Error Correction:** We execute the generated code and ask LLM to regenerate the reward function with the error trace if the code fails.

The generation process is carried out through a multi-round conversation with GPT-4o, following these steps:

1. **Staging:** In this step, the LLM is tasked with planning the reward function based on the preference. (Prompt 7)
2. **Coding:** The LLM is tasked with generating reward function code snippets according to its plan. (Prompt 8)
3. **Geometry Review:** The LLM reviews the generated code and corrects any geometry-related mistakes. (Prompt 9)
4. **Reward Normalization Review:** The LLM reviews and corrects normalization-related mistakes in the generated code. (Prompt 10)
5. **Code Cleanup:** The LLM reviews and resolves any coding or formatting errors in the generated code. (Prompt 11)
6. **Construct Reward Function:** The final reward function is constructed by inserting the generated code snippets into Template 12.

**Prompt used:**

> **Long Horizon Staging Prompt**
>
> ```
> # Instructions
>
> You are a reward engineer that is an expert at designing reward
> functions to solve reinforcement learning tasks. We have trained a
> reinforcement learning-based robot in a task environment. We then
> demonstate the policy to a human who gave feedback on the robot's
> performance. Now, we want to design a reward function that
> incorporates the human's feedback
> You will output an outline for the reward function `
> compute_dense_reward`, in natural language. It should be concrete
> and ready for implementation.
>
> You will be given:
> ```

```
1. `Original Code`: Code for a task environment class, including the
 reward function robot has already been trained in, namely `
compute_dense_reward`.
2. `Original Task Goal`: The original task goal of the robot before
the feedback was given.
3. `Simulation Environment Setup`: Description of the simulation
environment.
4. `Demonstration Summary`: Summary of the robot's performance
during the demonstration.
5. `Human Feedback`: Feedback given to the robot by a human who
watched it after it had been trained on the existing version of `
compute_dense_reward`.
6. `New Task Goal`: The new task goal for the robot after the
feedback has been incorporated.

You should:
1. Generate a high level plan for `compute_dense_reward`.
2. Design a staged reward:
  - Split the task into stages and give the agent reward gradually,
 encouraging it to complete each stage. The reward should therefore
 accumulate, NOT be all-or-nothing at the end.
  - Consider interdependencies and tradeoffs between different task
 stages and different aspects of the feedback and overall goal.
Ensure your reward design isn't counterproductive.
3. If there are aspects of the feedback that it's impossible to
incorporate without modifying other methods, say "I cannot do <
aspect>" and ONLY incorporate the other parts of the feedback, if
there are any.
4. If there are aspects of the feedback that are physically
impossible, say "I cannot do <aspect>" and ONLY incorporate the
other parts of the feedback, if there are any.
5. Explain your reasoning at each step.

# Design details
## `compute_dense_reward`

There are three stage templates you can use. You CANNOT use
anything else. The robot is ONLY capable of these three stages.

#### "pick up"
Template:
- Action: reward the robot for picking up <desired object>.
- Outcome: <desired object> is in the robot's gripper.
Example:
- Action: reward the robot for picking up `red_cube`.
- Outcome: `red_cube` is in the robot's gripper.

#### "place"
Template:
- Action: reward the robot for placing <desired object> <desired
position>.
- Outcome: <desired object> is <desired position>.
Example:
- Action: reward the robot for placing `red_cube` next to `sphereB`.
- Outcome: `red_cube` is next to `sphereB`.

#### "push"
Template:
- Action: reward the robot for pushing <desired object> to <desired
 position>.
- Outcome: <desired object> is <desired position>.
Example:
```

```markdown
- Action: reward the robot for pushing `red_cube` just left of the
center of the target.
- Outcome: `red_cube` is just left of the center of the target.

# Output format
Your output should be:
An ordered list of stage plans. Format:
```markdown
### Stage <stage number>: <language description of stage number>
- Stage template: "<pick up or place>"
  - **Action:** <language description of action>
  - **Outcome:** <language description of outcome>
- **Dependencies:** <reasoning about any dependencies with previous
 stages>
- **Other reasoning:** <whatever you think is important to note!>
```

Examples:
```markdown
### Stage 1: Pick up `obj1`
- Stage template: "pick up"
  - **Action:** Reward the robot for picking up `obj1`.
  - **Outcome:** `obj1` is in the robot's gripper.
- **Dependencies:** The robot can do this easily, so there aren't
prior dependencies.
- **Other reasoning:**: [open-ended, you will not be provided with
an example]
```

```markdown
### Stage 4: Put `obj2` next to `obj1`.
- Stage template: "place"
  - **Action:** Reward the robot for placing `obj2` next to `obj1`.
  - **Outcome:** `obj2` is next to `obj1`.
- **Dependencies:** Requires that the robot is grasping `obj2`.
- **Other reasoning:**: [open-ended, you will not be provided with
an example]
```

Notice that stage index starts from 0.

## Original Code:
```python
{ENVIRONMENT CODE}
```

## Original Task Goal: {CURRENT TASK GOAL}

## Simulation Environment Setup: {ENVIRONMENT SETUP DESCRIPTION}

## Demonstration Summary: {DEMO SUMMARY}

## Human Feedback: {GROUNDED FEEDBACK}

## New Task Goal: {NEW TASK GOAL}

Think step-by-step to make a high-level plan for writing `
compute_dense_reward`. The plan should be a series of stages.
- Don't code yet, just plan the stages.
- Consider dependencies and conflicts between task stages, and
create a plan that doesn't do anything counterproductive.

RULES YOU MUST FOLLOW:
```

```
1. Use ONLY the stage templates you've been given.
2. Incorporate human feedback as given.
3. Do not add anything that contradicts the feedback.
4. Do not invent your own changes or speculate about what the
feedback is going for.
5. Try accomplish the task goal with minimal stages. Long plans are
 harder to train on.
6. Make the plan complete - think about all the atomic stages the
robot needs to achieve every part of the feedback, not just the
final ones.
```

Long Horizon Coding Prompt

```
Excellent. Now write the relevant code for each stage.

# Instructions:
1. Don't add your own helper methods. Only edit what I've told you
to edit.
2. Go through the coding tips below.
3. Explain your implementation.
4. If there are aspects of the feedback that are physically
impossible, say "I cannot do <aspect>" and ONLY incorporate the
other parts of the feedback, if there are any.
5. Number of stages should match the number of stages in the plan.
It can be different from the number of stages in the original code.
# How to calculate target positions and position rewards

### Coding best practices
- Pay attention to the `Original Code` and the `Simulation
Environment Setup` given to you earlier.
- All positions are numpy arrays of shape (3,). Make sure you don't
 modify the content in info dictionaries.
= Don't worry about missing arguments

### How to set a target position for the robot to pick at/place at.
- Object coordinates are x, y, and z coordinates. **Explicitly
reason about every coordinate when setting the target position**.
- Note that those are x, y, z coordinates of the *center* of
objects.
  - So when setting target positions or getting object positions,
if you want the top, bottom, or sides of an object, add the right
offsets.
  - Otherwise, make sure you're reasoning about the center.
- Consider common sense physical issues. **Read the environment
code to understand geometry and dimension values.** Examples:
  - A box has walls, so if you want to put something inside the box,
 consider its wall width.
  - A target might be a circle or a square - if you want to put
something at the edge of the target, consider its shape.
- Remember that direction matters, not just distance. Let's say
your intended target position is two block side-lengths away from a
 block center. You can't take the block center, then tell the robot
 to go two block side-lengths away from that. It could go two block
 side-lengths in any direction! You have to calculate your intended
 position, unambiguously.

### How to write a reward component that gets the robot to choose
the right location
- Add a reward component that minimizes the difference between the
position the robot is currently planning to go to, and the target
position you set
```

```
  - Normalize and smooth these differences with a function like `np.
tanh`.
    - For example, if the robot has selected position `
current_selected_pos` and you set target position `target_pos`, you
can take the hyperbolic tangent of the norm of the difference
between these two.
    - You can also increase the coefficient on the norm within the `
np.tanh` to encourage `target_pos` more aggressively, you can
subtract from 1.0 so that the overall reward ranges from 0 to 1 and
 increases as distance decreases, and/or you can use a different
normalizing function based on the reward landscape you want.
    - **Design based on what we need from this reward component.**
- DON'T give a step function-like boost that only activates when
the object has reached its target position. If you do this, the
agent won't be able to get started.

# Documentation of function inputs
Functions take in two arguments from the environment: `prev_info`
and `cur_info`. These are dictionaries with the following keys:

{ENVIRONMENT INFORMATION KEYS}

`prev_info` represent the environment state BEFORE `action` was
taken and `cur_info` represent the state AFTER `action` was taken.

## Simulation Environment Setup: {ENVIRONMENT SETUP DESCRIPTION}

## Demonstration Summary: {DEMO SUMMARY}

## Human Feedback: {GROUNDED HUMAN FEEDBACK}

## New Task Goal: {NEW TASK GOAL}

# Your task:

Fill out the TODOs in this markdown to get reward. Fill out a new
copy of the markdown for every stage. Write a short chain of
reasoning before each code block to explain your reasoning.

```markdown
stage N target action: # TODO: select [`pick`, `place`, `push`].
Select EXACTLY the one given to you in the plan - DO NOT make your
own judgement call.

```python
def compute_target_position_stageN(self, prev_info, cur_info):
    """
    Defines the target position for the robot's action - the
location it needs to pick at/place at/push to.
    Has no return. The function ends in the definition of `
target_pos_1` and `target_pos_2`.

    Arguments:
        - `self (BaseEnv)`: gives access to environment attributes
and method calls.
        - `prev_info (dict[str, Any])`: state representation of the
environment state BEFORE `action` was taken.
        - `cur_info (dict[str, Any])`: same as `prev_info`, except
from the state AFTER `action` was taken.
    """
    # TODO: implement target position for stageN's target action
based on environment.
```

```
    # If the action is `pick` or `place`, target_pos_1 must be a 3D
position indicating the target position for the action, and
target_pos_2 must be None.
    # If the action is `push`, target_pos_1 must be a 3D position
indicating the target starting position for the push action, and
target_pos_2 must be a 3D position indicating the target ending
position for the push action.
    target_pos_1 = ...
    target_pos_2 = ...
```

```python
def compute_target_pos_reward_stageN(self, prev_info, cur_info,
current_selected_pos_1, current_selected_pos_2, target_pos_1,
target_pos_2):
    """
    Sets a reward to encourage the robot to take the action at `
target_pos_1` and `target_pos_2`. This reward should be dense,
setting a continuous-valued penalty for any difference between `
current_selected_pos_1` and `current_selected_pos_2` (the location
the robot IS CURRENTLY PLANNING to pick at/place at/push to) and `
target_pos_1` and `target_pos_2` (the location the robot SHOULD pick
 at/place at/push to).
    Has no return. The function ends in the definition of `reward`.

    Arguments:
        - `self (BaseEnv)`: gives access to environment attributes
and method calls.
        - `prev_info (dict[str, Any])`: state representation of the
environment state BEFORE `action` was taken.
        - `cur_info (dict[str, Any])`: same as `prev_info`, except
from the state AFTER `action` was taken.
        - `current_selected_pos_1`  and `current_selected_pos_2` (
numpy.ndarray): the pos the robot IS CURRENTLY PLANNING TO take the
 action at
        - `target_pos_1 (numpy.ndarray)` and `target_pos_2 (numpy.
ndarray)`: `target_pos` defined in `compute_target_position_stageN`
- the pos the robot SHOULD take the action at

    Notice that current_selected_pos_2 and target_pos_2 are None if
 the action is `pick` or `place`. Do not use them in this case.
    This function will only be called if selected action equals
target action. In other words, current_selected_pos_2 and
target_pos_2 will be both None or both numpy arrays.
    """
    reward = # TODO implement a *dense and normalized* reward for
guiding the robot to do `target_action` at `target_pos`s, i.e. align
 `current_selected_pos_1` and `current_selected_pos_2` with `
target_pos_1` and `target_pos_2`. Use the `target_pos_1` and `
target_pos_2` param, don't calculate your own.
    # - Feel free to use normalization functions like `np.tanh`.
Your reward MUST be normalized to between -1 and 0.
    # - Ensure your reward is *dense*. Do not give a sudden boost
for reaching the target position. Instead, make sure your reward
implementation gradually and continuously guides the robot to
target positions. Otherwise, the robot won't be able to get started.

```

```python
def stageN_success(info):
    """
```

```
    Return true if the robot has successfully completed stageN,
else return false.

    Arguments:
        - `info (dict[str, Any])`: state representation of the
current environment state.

    Returns:
        - `bool`: True if the robot has successfully completed
stageN, else False.
    """
    # TODO
```
```

Don't write `compute_dense_reward` as a whole. Only write the
functions I asked for.
```

---

**Long Horizon Reward Function Geometry Review Prompt**

```
Review the setup description again:

{ENVIRONMENT SETUP DESCRIPTION}

- For every target position, did you specify all three coordinates?
- Did you set target positions **exactly where you desire the robot
 to go**? Common pitfalls:
- Adding a small error threshold to the target position. Always
allow a bit of error when calculating a *boolean check* in `
stageN_success`, but the target positions themselves should have no
 error.Âă
- Did you reason about the dimensions of each object and how they
might impact target positions?
- Picking an object up requires positioning the gripper around the
center of the object. Did you consider this? In other words, did
you set the target position to the center of the object?

Verify. If you need to make any edits, do so.
1. Only output functions you are editing.
2. If you are editing a function, output the whole thing. Only do
so once, with all corrections made.
```

---

**Long Horizon Reward Normalization Review Prompt**

```
- Did you consider the range of your normalization functions
correctly? Common pitfalls: make sure the range of the reward is -1
 to 0.
- Did you consider sensitivity thoughtfully? Common pitfalls: not
having tolerance for small errors in the reward and success
calculation.

Verify. If you need to make any edits, do so.
1. Only output functions you are editing.
2. If you are editing a function, output the whole thing. Only do
so once, with all corrections made.
```

**Long Horizon Reward Function Cleanup Prompt**

```
- Did you make sure not to include return statements in all `
compute_target_position_stageN` and `
compute_target_pos_reward_stageN`, and instead just define `
target_pos` and `reward`, respectively?
- Did you set any new instance variables, i.e. did you create any
new `self.<varname>`? You should not - you should just set `
target_pos` and `reward` as regular variables. I will handle the
rest.
- Did you use the right datatypes? All positions should be numpy
arrays of shape (3,), rewards should be floats, and `stageN_success`
 returns boolean.

Verify. If you need to make any edits, do so.
1. Only output functions you are editing.
2. If you are editing a function, output the whole thing. Only do
so once, with all corrections made.
```

**Long Horizon Reward Function Template**

```python
def evaluate(self):
    info = self._get_obs_info()

    def stage0_success(info):
        <GENERATED STAGE 0 SUCCESS CONDITION>
    ...

    def stageN_success(info):
        <GENERATED STAGE N SUCCESS CONDITION>

    info["stage0_success"] = stage0_success(info)
    ...
    info["stageN_success"] = stageN_success(info)

    info["success"] = torch.tensor(False)
    if self.cur_stage==N:
        info["success"] = torch.tensor(info["stageN_success"])

    return info

def compute_dense_reward(self, prev_info, cur_info, action, **
kwargs):
    reward_components = dict((k, 0.0) for k in self.
reward_components)
    current_selected_action = np.argmax(action[:len(self.
task_skill_indices.keys())])

    if current_selected_action in [0, 1]:
        current_selected_pos_1 = action[len(self.task_skill_indices.
keys()):len(self.task_skill_indices.keys())+3]
        current_selected_pos_2 = None
    else:
        current_selected_pos_1 = action[len(self.task_skill_indices.
keys()):len(self.task_skill_indices.keys())+3]
        current_selected_pos_2 = action[len(self.task_skill_indices.
keys())+3:len(self.task_skill_indices.keys())+6]

     def stage_0_reward():
        <GENERATED TARGET POSITION AND ACTION DEFINITION>

        if current_selected_action == target_action:
```

```
            <GENERATED DISTANCE DEFINITION>

            reward_components["afford"] = (1 + reward) * 5.0

        if cur_info["stage0_success"]:
            reward_components["success"] = 10.0
        return reward_components

    ...

    if self.cur_stage==0:
        reward = stage_0_reward()
    ...
    if self.cur_stage==N:
        reward = stage_N_reward()

    if (self.cur_stage == 0) and cur_info["stage0_success"]:
        self.cur_stage = 1
    ...
    if (self.cur_stage == N-1) and cur_info["stage{N-1}_success"]:
        self.cur_stage = N
    return reward
```

### C.3   PREFERENCE GROUNDING PROMPT

Feedback grounding consists of two stages: (1) translating a demonstration video and its trajectory into a language-based description, and (2) leveraging the language description and human feedback to generate grounded feedback and update task goals.

#### C.3.1   DEMONSTRATION TO LANGUAGE DESCRIPTION

To address the Vision-Language Action (VLA) model's limitations in spatial and motion understanding, we process each demonstration by decomposing it into frames and augmenting them with state information, including object 3D positions and gripper states.

**Short-Horizon Descriptions**    We process short-horizon descriptions in the following steps:

1. Sample frames at every 5-frame interval.
2. For each sampled frame, extract object positions, states, and the corresponding video frame, then prompt GPT-4o with Prompt 13 to generate a state description.
3. For the transitions between sampled frames, input the consecutive state descriptions into GPT-4o with Prompt 15 and request a description of the motion occurring between them.
4. Compile all state and motion descriptions into an interleaved sequence of states and actions.

**Long-Horizon Descriptions**    For long-horizon descriptions, we segment the demonstration into a sequence of primitive actions and process them as follows:

1. Generate a separate language description for each action's start and end state by providing GPT with the corresponding video frame and state information using Prompt 14.
2. Provide the start state description, end state description, primitive action type, and action parameters as input to GPT-4o with Prompt 16, requesting a language-based description of the action.
3. Compile all state and motion descriptions into an interleaved sequence of states and actions.

**Prompts used:**

Short Horizon State Description Prompt

```
Given the image from a robotic simulation, a description of the
setup, and state information, write a caption describing the scene.
## Your response should be similar to the following examples:
EXAMPLE 1: A robot gripper is slightly above the sphere. The sphere
 is on the table, next to the bin.
EXAMPLE 2: The robot is holding a blue cube above the orange cube.
The orange cube is on stacked on top of the pink cube.
## Input
Setup Description: {ENVIRONMENT SETUP DESCRIPTION}
Here is the state information:  {STATE}
Objects do not float. If an object is elevated, it is either held
by the robot, stacked on top of another object, or inside a drawer
or a bin.

<ENCODED FRAME>
```

Long Horizon State Description Prompt

```
Given the image from a robotic simulation, a description of the
setup, and state information, write a caption describing the scene.
## Your response should be similar to the following examples:
EXAMPLE 1: A robot gripper hovers above a wooden table, holding a
green cube, while a red cube and a purple ball rest on the table
surface.
EXAMPLE 2: The robot is holding a green ball above the table. There
 are two cubes stacked together on the table. The red cube is on
top of
the purple cube.
## Input
Setup Description: {ENVIRONMENT SETUP DESCRIPTION}
Here is the state information:  {STATE}
Objects do not float. If an object is elevated, it is either held
by the robot, stacked on top of another object, or inside a drawer
or a bin.

<ENCODED FRAME>
```

Short Horizon Action Description Prompt

```
Your job is to generate a language description of a robot's action
based on
(1) START STATE: a dictionary containing the state and position of
objects before the action,
(2) START STATE DESCRIPTION: the language description of the start
state,
(3) END STATE: a dictionary containing the state and position of
objects after the action,
(4) END STATE DESCRIPTION: the language description of the end
state, and "
Environment description: {ENVIRONMENT SETUP DESCRIPTION}
EXAMPLE 1: The robot released the blue cube on top of the green
cube,
EXAMPLE 2: The robot picks up the blue cube.

Here are the inputs
START STATE: {START STATE}
START STATE DESCRIPTION: {START STATE LANGUAGE DESCRIPTION}
END STATE: {END STATE}
```

```
END STATE DESCRIPTION: {END STATE LANGUAGE DESCRIPTION}
Now generate a description of the robot's action. Focus on the
coordinates. State description might be wrong.
```

**Long Horizon Action Description Prompt**

```
Your job is to generate a language description of a robot's action
based on

(1) START STATE: a dictionary containing the state and position of
objects before the action,
(2) START STATE DESCRIPTION: the language description of the start
state,
(3) END STATE: a dictionary containing the state and position of
objects after the action,
(4) END STATE DESCRIPTION: the language description of the end
state, and
(5) ACTION DESCRIPTION: the parameter of the action command to the
robot.
Environment description: {ENVIRONMENT SETUP DESCRIPTION}

Your response should be one or two sentences long and should be
similar to the following examples:

EXAMPLE 1: The robot successfully picks up the green cube.
EXAMPLE 2: The robot trys to place down the red cube on the green
cube. However, the red cube ended on the table.

Here are the inputs
START STATE: {START STATE}
START STATE DESCRIPTION: {START STATE LANGUAGE DESCRIPTION}
END STATE: {END STATE}
END STATE DESCRIPTION: {END STATE LANGUAGE DESCRIPTION}
ACTION DESCRIPTION: {ACTION}

Now generate a description of the robot's action.
```

### C.3.2 GROUNDED PREFERENCE AND TASK GOAL GENERATION

With the generated language description of the demonstration, we employ Prompt 17 to generate grounded preference and language summary of the demonstration and Prompt 18 to update the task goals.

**Prompts used:**

**Grounded Preference Prompt**

```
You are a helpful assistant. You are skilled at taking things
people say off the cuff and without context, and specifying exactly
 what they mean in unambiguous, clear language.

I showed a person a video of a robot doing a task and asked them to
 give feedback to/about the robot. You will be given:
1. Description of the task
2. Frame-by-frame description of the video
3. The person's feedback - could be ambiguous, confusing, overly
long, overly short, full of errors, etc.
```

```
You need to provide:
1. A summary of the video - descriptive, complete, concise.
2. A rewrite of the feedback that is unambiguous, contextualizes
the person's feedback in the video/task description, uses standard
language, and directs the robot to do exactly what the person *
meant* to ask for.

# Feedback grounding examples
UNGROUNDED FEEDBACK 1: Not that one!
GROUNDED FEEDBACK 1: The robot moved Blue Cube to the right. It
should not - it should move Red Cube to the right.
EXPLANATION 1: The person says "not that one" and there are two
cubes in the scene, meaning they appear to have liked the action,
but not the specific cube it was being done on. So, I added in
direct references to the Blue Cube and the Red Cube. Since it was
the Blue Cube that was moved to the right and the person clearly
wanted the other option, I said not to do that and to instead move
the Red Cube to the right.

UNGROUNDED FEEDBACK 2: Hmmm, I like that the it move arm like that
and put banana smoothly? but can it put banana in another one?
GROUNDED FEEDBACK 2: Arm movement is good, but put Banana in Shelf
2, Shelf 3, or Shelf 4.
EXPLANATION 2: The person likes the arm movement. They want the
banana to be put in another "one" - since it was previously put on
a shelf, this means they want it put in a different shelf.
Previously it was shelf 1, so now I ask for Shelves 2, 3, or 4.
They also liked the way the robot put the Banana in Shelf 1, but
now they're asking for it to be put in a different one, so even
though they liked it previously, I don't ask for that now.

UNGROUNDED FEEDBACK 3: the block can go further
GROUNDED FEEDBACK 3: Move Cube 1 further to the right.
EXPLANATION 3: The person referenced a "block". There's nothing in
the description called a "block", but there is a "cube", which is a
 synonym. They want it moved further but don't specify where, which
 suggests they want it moved further in the original direction.
That was to the right, so I preserve that here.

# Task description
{CURRENT TASK GOAL}

# Video description
{LANGUAGE DESCRIPTION OF THE DEMO}

# Feedback
"{RAW HUMAN FEEDBACK}"

# Your task
1. Think step by step to briefly summarize the demonstration, then
go through the checklist, then give me your final version of the
feedback, grounded to the description.
2. Make sure your grounded feedback has a directive tone.
3. Use standard, unambiguous language over using the person's own
language.
4. Do not have ANY ambiguous objects. ALWAYS use the official term
for an object if it has one.

CHECKLIST:
- The feedback may not be ordered, even if it contains multiple
parts. Did you assume order inappropriately?
- The person might compliment something about the past. Don't
conflate that with what they're asking for *now*. Did you make sure
```

```
 to isolate only the current instructions, and remove things they
like but don't want to keep?
- The person will NEVER reference something not in the video. If
they use a word you don't think is in the video, YOU ARE WRONG. Did
 you figure out what they were talking about? Did you replace any
unexpected terms with official terms from the description?

# Output
JSON format:
{
    "summary": "<SUMMARY>",
    "explanation": "<EXPLANATION>",
    "feedback": "<GROUNDED FEEDBACK>",
}
```

Task Goal Update Prompt

```
You are a helpful assistant that is excellent at interpreting what
humans are asking for. You will be shown three phrases:
1. A task description that was attempted by a robot
2. A description of the robot's attempted demonstration
3. Original feedback the person gave after watching the robot make
the attempt.
4. Grounded feedback that has been rewritten to be clear and
unambiguous.

Your job:
- If the person's feedback changes the task itself such that the
original task description no longer applies, rewrite the task
description.
- If the person's feedback adds to the task, then add the new
information to the task description.
- However, if the person's feedback doesn't change the task and
only comments on the robot's execution, then output an empty string.

- Give these in a json output.

Current task description: {task_description}
Demo description: {demo_description}
Original Feedback: {original_feedback}
Grounded Feedback:  {grounded_feedback}

# Your output should be a JSON object with the following format:
{
    "task_description": <TASK DESCRIPTION>,
}
```

## C.4 ENVIRONMENT SPECIFIC PROMPT

### C.4.1 PUSHBALL

PushBall Setup Description

```
There should be a robot gripper and a ball on the table. In the 3D
coordinate of the projects [x,y,z], x represents left and right
positive is left, negative is right, y represents forward and
backward, positive is forward, negative is backward, and z
represents height. z = 0 represents the table surface. The radius
```

of the sphere is 0.06. Position is measured at each object's center.
 Expect errors in the measurement.

### C.4.2 PLACESPHERE2BINWIDE

**PlaceSphere2BinWide Setup Description**

There should be a robot gripper, a sphere, and two bins on the
table. In the 3D coordinate of the projects [x,y,z], x represents
forward and backward, positive is forward, y represents left and
right positive is left, negative is right,  negative is backward,
and z represents height. z = 0 represents the table surface. The
radius of the sphere is 0.02 and each bin is 0.16 wide and 0.01
deep. Position is measured at each object's center. Expect errors
in the measurement.

### C.4.3 STACK3CUBE

**Stack3Cube Setup Description**

There are a robot gripper, a red cube, a green cube, and a purple
cube on a table. The 3D coordinates of the projects [x,y,z] are
defined from the viewer's perspective: the x-axis represents
forward and backward, with positive values being closer and
negative values farther away from the viewer; the y-axis denotes
horizontal direction, with negative values to the left and positive
 to the right of the viewer. The z-axis measures height, where z=0
corresponds to the table surface. Each cube is 0.04 by 0.04 by 0.04.
 Position is measured at each object's center. Expect errors in the
 measurement.

**Stack3Cube Information Keys**

- `red_cube_pos`: 3D coordinate of red_cube
- `green_cube_pos`: 3D coordinate of green_cube
- `purple_cube_pos`: 3D coordinate of purple_cube
- `is_red_cube_grasped`: whether red_cube is grasped by the robot
- `is_green_cube_grasped`: whether green_cube is grasped by the
robot
- `is_purple_cube_grasped`: whether purple_cube is grasped by the
robot
- `gripper_pos`: 3D coordinate of the robot's gripper

### C.4.4 PUTOBJECTINDRAWER

**PutObjectInDrawer Setup Description**

There should be a robot gripper, a red apple, a red soup can, and a
 drawer on the floor. Objects are roughly 0.05 in height and width.
 Object positions are measured at the center of the object. The 3D
coordinates of the projects [x,y,z] are defined from the viewer's
perspective: the x-axis represents forward and backward, with
positive values being closer and negative values farther away from
the viewer; the y-axis denotes horizontal direction, with negative
values to the left and positive to the right of the viewer. The z-

```
axis measures height, where z=0 corresponds to the floor. The
bottom drawer is facing the right (+y direction). It's open when
the scene starts. The drawer is 0.36 wide, 0.22 deep, and 0.16 high.
 In other words, x = drawer_pos[0] + 0.18 is the left edge of the
drawer, x = drawer_pos[0] - 0.18 is the right edge of the drawer, y
 = drawer_pos[1] + 0.11 is the drawer front, z = drawer_pos[2] +
0.16 is the top of the drawer, and z = drawer_pos[2] is the bottom
of the drawer. The robot is placed on the floor and cannot reach
the inside of the drawer. To put an object in the drawer it must be
 dropped from above.
```

**PutObjectInDrawer Information Keys**

```
- `apple_pos`: 3D coordinate of apple
- `soup_can_pos`: 3D coordinate of soup_can
- `is_apple_grasped`: whether apple is grasped by the robot
- `is_soup_can_grasped`: whether soup_can is grasped by the robot
- `gripper_pos`: 3D coordinate of the robot's gripper
- `drawer_handle_pos`: 3D coordinate of the bottom drawer handle
- `drawer_pos`: 3D coordinate of the center bottom of the drawer
- `drawer_open_offset`: how much the drawer is open, measured as the
 distance between the drawer's position now and when it is fully
closed
```

### C.4.5 OBJECTTOBIN

**ObjectToBin Setup Description**

```
There should be a robot gripper, an apple, an orange, a baseball, a
 tennis ball, and two bins on the table. One bin is light blue and
one is white. The 3D coordinates of the projects [x,y,z] are
defined from the viewer's perspective: the x-axis represents
forward and backward, with positive values being closer and
negative values farther away from the viewer; the y-axis denotes
horizontal direction, with negative values to the left and positive
 to the right of the viewer. The z-axis measures height, where z=0
corresponds to the table surface. Each object is about 0.05 in
diameter. Each bins are 0.20 by 0.20 by 0.02 in size. Position is
measured at each object's center. Expect errors in the measurement.
```

**ObjectToBin Information Keys**

```
- `apple_pos`: 3D coordinate of apple
- `orange_pos`: 3D coordinate of orange
- `baseball_pos`: 3D coordinate of baseball
- `tennis_ball_pos`: 3D coordinate of tennis_ball
- `blue_bin_pos`: 3D coordinate of blue_bin
- `white_bin_pos`: 3D coordinate of white_bin
- `is_apple_grasped`: whether apple is grasped by the robot
- `is_orange_grasped`: whether orange is grasped by the robot
- `is_baseball_grasped`: whether baseball is grasped by the robot
- `is_tennis_ball_grasped`: whether tennis_ball is grasped by the
robot
- `gripper_pos`: 3D coordinate of the robot's gripper
```

## C.5 BASELINES PROMPTS

---

**Eureka Prompt Template**

```
We are going to use a Franka Panda robot to complete given tasks.
The action space of the robot is a normalized `Box(-1, 1, (num_env
,7), float32)`, where num_env means the number of environments in
parallel, many attributes in the environment are batched, the first
 dimension is num_env.

# Environment Code:
{environment_code}

# Maniskill Doc:
{reward_guidelines}

# Previous Implementations:

Previous reward function:
```python
{previous_reward}
```

Your reward function code has been analyzed, the feedback is as
follows:
# Human Feedback:
{human_feedback}

# Objective Feedback:
{objective_feedback}

Please carefully analyze the policy feedback and provide a new,
improved reward function that can better solve the task. Some
helpful tips for analyzing the policy feedback:
    (1) If the success rates are always near zero, then you must
rewrite the entire reward function

Please generate a reward function that follows all guidelines and
addresses the human feedback. The code output should be formatted
as a python code string: "```python ... ```".
- "compute_dense_reward": containing the complete reward function
code

The functions MUST have these EXACT signatures:

def compute_dense_reward(self: BaseEnv, obs: Any, action: torch.
Tensor, info: Dict[str, torch.Tensor]) -> torch.Tensor
    #Encodes reward for each possible action based on `evaluate`
output dictionary and other current environment info (environment
instance attributes).

    #Incorporates human feedback as given.

    #Obeys the following structure:
    #1. Stage reward
      - Task is split into stages and reward is given to the agent
gradually, encouraging it to complete each stage. The reward
accumulates.
      - Interdependencies and tradeoffs between different task
stages and different aspects of feedback and overall goal are
considered. Reward design is not overall counterproductive to meet
short-term goals.
    #2. Return reward value
```

```
    # Your implementation here
    pass
```

The function should:
1. Match the exact function signatures shown above
2. Handle batched operations correctly (num_env dimension)
3. Include comprehensive comments
4. Follow the reward function best practices
5. Be consistent with the interface of the previous implementations
 while incorporating the requested changes

The code output should be formatted as a python code string: "```
python ... ```".

---

**Text2Reward Prompt Template**

We are going to use a Franka Panda robot to complete given tasks.
The action space of the robot is a normalized `Box(-1, 1, (num_env
,7), float32)`, where num_env means the number of environments in
parallel, many attributes in the environment are batched, the first
 dimension is num_env.

# Task-Specific Environment:
{environment_description}

# Available Objects and Their Properties:
{env_class_desc}

# Maniskill Doc:
{reward_guidelines}

# Previous Implementations:

Previous reward function:
```python
{previous_reward}
```

# Human Feedback:
{human_feedback}

Please generate a reward function that follows all guidelines and
addresses the human feedback. The code output should be formatted
as a python code string: "```python ... ```".
- "compute_dense_reward": containing the complete reward function
code

The functions MUST have these EXACT signatures:

def compute_dense_reward(self: BaseEnv, obs: Any, action: torch.
Tensor, info: Dict[str, torch.Tensor]) -> torch.Tensor
    #Encodes reward for each possible action based on `evaluate`
output dictionary and other current environment info (environment
instance attributes).

    #Incorporates human feedback as given.

    #Obeys the following structure:
    #1. Stage reward

```
        - Task is split into stages and reward is given to the agent
gradually, encouraging it to complete each stage. The reward
accumulates.
        - Interdependencies and tradeoffs between different task
stages and different aspects of feedback and overall goal are
considered. Reward design is not overall counterproductive to meet
short-term goals.
    #2. Return reward value
    # Your implementation here
    pass
```

The function should:
1. Match the exact function signatures shown above
2. Handle batched operations correctly (num_env dimension)
3. Include comprehensive comments
4. Follow the reward function best practices
5. Be consistent with the interface of the previous implementations
 while incorporating the requested changes

The code output should be formatted as a python code string: "```
python ... ```".
```

---

**Text2Reward Environment Class Discription**

```
#class PandaRobot:
    self.tcp.pose.p: torch.Tensor([num_env, 3]) # Batched by the
number of environment, indicate the 3D position of the robot's
gripper
    self.tcp.pose.q: torch.Tensor([num_env, 4]) # Batched by the
number of environment, indicate the quaternion of the robot's
gripper
    self.robot.qpos: torch.Tensor([num_env, 9]) # Batched by the
number of environment, indicate the joint positions (last 2 for
gripper) of the robot at this key frame
    self.robot.qvel: torch.Tensor([num_env, 9]) # Batched by the
number of environment, indicate the joint velocities (last 2 for
gripper) of the robot at this key frame
    def is_grasping(self, object: Actor, min_force=0.5, max_angle
=85): -> torch.Tensor([num_env, ], bool) # Batched by the number of
 environment, check if the robot is grasping an object

class RigidObject:
    self.pose.p: torch.Tensor([num_env, 3]) # Batched by the number
 of environment, indicate the 3D position of the simple rigid
object in each environment
    self.pose.q: torch.Tensor([num_env, 4]) # Batched by the number
 of environment, indicate the quaternion of the simple rigid object
 in each environment
    def get_angular_velocity(self,) -> torch.Tensor([num_env, 3]) #
 Batched by the number of environment, indicate the angular
velocity of the simple rigid object
    def get_linear_velocity(self,) -> torch.Tensor([num_env, 3]) #
Batched by the number of environment, indicate the linear velocity
of the simple rigid object
    self.get_first_collision_mesh().bounding_box.bounds: np.ndarray
[(2, 3)] # non-batched, indicate the bounding box of the simple
rigid object

class TargetObject:
```

```
    self.pose.p: torch.Tensor([num_env, 3]) # Batched by the number
 of environment, indicate the 3D position of the TargetObject in
each environment
    self.pose.q: torch.Tensor([num_env, 4]) # Batched by the number
 of environment, indicate the quaternion of the TargetObject in
each environment

class LinkObject:
    self.pose.p: torch.Tensor([num_env, 3]) # Batched by the number
 of environment, indicate the 3D position of the link object in
each environment
    self.pose.q: torch.Tensor([num_env, 4]) # Batched by the number
 of environment, indicate the quaternion of the link object in each
 environment
    def get_angular_velocity(self,) -> torch.Tensor([num_env, 3]) #
 Batched by the number of environment, indicate the angular
velocity of the link object
    def get_linear_velocity(self,) -> torch.Tensor([num_env, 3]) #
Batched by the number of environment, indicate the linear velocity
of the link object
    self.qpos : torch.Tensor([num_env,],float) # Batched by the
number of environment, indicate the position of the link object
joint
    self.qvel : torch.Tensor([num_env,],float) # Batched by the
number of environment, indicate the velocity of the link object
joint

class ArticulateObject:
    self.pose.p: torch.Tensor([num_env, 3]) # Batched by the number
 of environment, indicate the 3D position of the articulate object
in each environment
    self.pose.q: torch.Tensor([num_env, 4]) # Batched by the number
 of environment, indicate the quaternion of the articulate object
in each environment
    self.qpos : torch.Tensor([num_env, 9]) # Batched by the number
of environment, indicate the joint positions of the articulate
object at this key frame
    self.qvel : torch.Tensor([num_env, 9]) # Batched by the number
of environment, indicate the joint velocities of the articulate
object at this key frame
    self.get_first_collision_mesh().bounding_box.bounds: np.ndarray
[(2, 3)] # non-batched, indicate the bounding box of the articulate
 object
```

---

**Documentation For Eureka and Text2Reward**

```
    # Documentation of classes, methods, and properties you can use
 when implementing `compute_dense_reward`

This is a parallel training environment with many episodes. Each
pose, boolean description of the scene, etc. is therefore a `torch.
Tensor` where the first dimension is the *batch dimension*. Each
episode is independent, so you must calculate reward separately for
 each one. Specifically:
  - `reward` is a `torch.Tensor` of scalar reward values for the
batch of episodes. The first dimension is a batch of episodes, and
the second dimension is the scalar reward value for that individual
 episode.
  - For example, if you want to give a reward boost of 5.0 to
exactly the episodes where the object is grasped, and you have
```

```
already computed the `is_obj_grasped` tensor, you can say `reward["
is_obj_grasped"] += 5.0`.

## `Actor` class
Non-robot objects in scene are subclasses of `Actor`.
### Properties
- `px_body_type`: `Literal["kinematic", "static", "dynamic"]`
indicating physics behavior - static (immovable), dynamic (physics-
affected), kinematic (not physics-affected).
  - The robot cannot move "static" and "kinematic" objects. If a
feedback asks you to change their locations, reject that part.
- `name`: `str` acting as the unique identifier for the actor
- `merged`: `bool` indicating if the actor is a composite of
multiple other actors
- `pose`: `Pose` object representing the current state
    - `Pose` object: dataclass
        - `pose.p` is a position
        - `pose.q` is a quaternion indicating orientation
        - `pose.raw_pose` is concatenation of `p` then `q`
        - `Pose.create_from_pq(p=p, q=q)` returns a new `Pose`
object at that `p` and `q`.
### Instance methods (called with `<actor_instance>.<method_name>(<
params>)`)
- `get_state`:
  - `self.<actor_instance>.get_state()[:, 7:10]`: contains linear
velocity
  - `self.<actor_instance>.get_state()[:, 10:13]`: contains angular
velocity
- `is_static(lin_thresh=1e-2, ang_thresh=1e-1)`: boolean check if
actor is static based on velocity thresholds
  - `lin_thresh`: linear velocity threshold
  - `ang_thresh`: angular velocity threshold

## Agent class
`BaseAgent` class for the robot. The robot is a 7-DoF arm-and-two
finger gripper. Degrees of freedom: arm position, arm orientation,
and gripper open/close.
- Note that you will see `tcp` throughout the code, with its
specific position being queried. `tcp` stands for Tool Center Point,
 the center between the two fingers. `tcp.pose` attributes tells you
 where the gripper is positioned.
### Properties
- `self.agent.controller`: currently activated controller
- `self.agent.action_space`: position/orientation/state that
controller has been sent to in the latest action, concatenated for
the two controllers (arm and end-effector)
### Methods
- `self.agent.get_state()`: returns a dictionary with the following
keys:
  - "robot_root_pose": root pose
  - "robot_root_vel": root velocity
  - "robot_root_qvel": root angular velocity
  - "robot_qpos": joint position
  - "robot_qvel": joint velocity
  - "controller": output of `controller.get_state()`, which contains
 target positions and orientations of the currently activated
controller
- `self.agent.is_grasping(object: Union[Actor, None] = None)`:
boolean check if agent is grasping object
- `is_static(threshold: float)`: boolean check if robot is static (
within given threshold) in terms of q velocity
- `self.agent.robot.get_qlimits()[0, -1, 1] * 2`: gripper width, i.e
. max possible opening
```

```
## Other
- `tcp` stands for "tool center point" - it is the point between the
  robot's grippers and is TODO whole robot
- Like `reward` and the values of `info`, quantities are tensors
over the batch. For example, `pose` is a tensor where the first
dimension is the batch of multiple episodes, and the rest describe
the pose of that episode.
- IMPORTANT: here, x is the front-and-back axis, y is the left-and-
right axis, and z is the up-and-down axis. Typically, x and y are
flipped - check your work!
```

### LMPC Prompt Template

```
## High-Level Description:
You are an expert robot reward function programmer.
Your goal is to write improved reward functions for a Franka Panda
robot arm with a gripper to fulfill tasks based on feedback and
previous implementations.
You will analyze feedback and create reward functions that better
guide the robot to accomplish its tasks.

## Robot Environment Details:
We are using a Franka Panda robot to complete given tasks. The
action space of the robot is a normalized `Box(-1, 1, (num_env,7),
float32)`, where `num_env` means the number of environments in
parallel. Many attributes in the environment are batched, with the
first dimension being `num_env`.

The robot is a 7-DoF arm with a two-finger gripper. Degrees of
freedom include arm position, arm orientation, and gripper open/
close. The Tool Center Point (tcp) is the center between the two
fingers.

## Coordinate System:
The coordinate system is right-handed and three-dimensional:
- x-axis: front-and-back
- y-axis: left-and-right
- z-axis: up-and-down (aligned with gravity)

Note that x and y are typically flipped in standard notation -
check your work carefully!

## Previous Implementation and Feedback:
For each task, you will be provided with:
1. A previous reward function implementation (`compute_dense_reward
`)
2. Human feedback on the previous implementation (`human_feedback`)

## Your Task:
You must carefully analyze the feedback and provide a new, improved
 reward function that better solves the task. Your function should:

1. Match the exact function signature:
```
def compute_dense_reward(self: BaseEnv, obs: Any, action: torch.
Tensor, info: Dict[str, torch.Tensor]) -> torch.Tensor
```

## Available APIs:
```

```
### Actor Class (Non-robot objects in scene)
#### Properties:
- `px_body_type`: Physics behavior type ("kinematic", "static", "
dynamic")
- `name`: Unique identifier
- `merged`: Boolean indicating if the actor is composite
- `pose`: Position and orientation
  - `pose.p`: Position
  - `pose.q`: Quaternion orientation
  - `pose.raw_pose`: Concatenation of p and q

#### Methods:
- `get_state()[:, 7:10]`: Linear velocity
- `get_state()[:, 10:13]`: Angular velocity
- `is_static(lin_thresh=1e-2, ang_thresh=1e-1)`: Boolean check if
actor is static

### Agent Class (Robot)
#### Properties:
- `self.agent.controller`: Currently activated controller
- `self.agent.action_space`: Position/orientation/state of
controllers

#### Methods:
- `self.agent.get_state()`: Returns robot state dictionary
- `self.agent.is_grasping(object)`: Boolean check if agent is
grasping object
- `is_static(threshold)`: Boolean check if robot is static
- `self.agent.robot.get_qlimits()[0, -1, 1] * 2`: Gripper width

### Important Notes:
- `tcp` is the Tool Center Point (center between robot's grippers)
- All quantities are tensors over the batch dimension
- The robot cannot move "static" and "kinematic" objects

## Output Format:
Your output should be formatted as a Python code string: "```python
... ```"
- The code should contain the complete `compute_dense_reward`
function

## Robot Code Writing Hints:
- Do not use any functions or object names besides the ones
mentioned above.

# Chat Turn Example:

## Environment code:

```python
class CubeAndTargetEnv(BaseEnv):
    """
    **Task Description:**
    A simple task where the objective is to pull a cube onto a
target.

    **Randomizations:**
    - the cube's xy position is randomized on top of a table in the
 region [0.1, 0.1] x [-0.1, -0.1].
    - the target goal region is marked by a red and white target.
The position of the target is fixed to be the cube's xy position -
[0.1 + goal_radius, 0]
```

```
    **Success Conditions:**
    - the cube's xy position is within goal_radius (default 0.1) of
 the target's xy position by euclidean distance.
    """

    _sample_video_link = "https://github.com/haosulab/ManiSkill/raw/
main/figures/environment_demos/PullCube-v1_rt.mp4"
    SUPPORTED_ROBOTS = ["panda", "fetch"]
    agent: Union[Panda, Fetch]
    goal_radius = 0.1
    cube_half_size = 0.02

    def __init__(self, *args, robot_uids="panda",
robot_init_qpos_noise=0.02, **kwargs):
        self.robot_init_qpos_noise = robot_init_qpos_noise
        super().__init__(*args, robot_uids=robot_uids, **kwargs)

    @property
    def _default_sensor_configs(self):
        pose = look_at(eye=[0.3, 0, 0.6], target=[-0.1, 0, 0.1])
        return [CameraConfig("base_camera", pose, 128, 128, np.pi /
 2, 0.01, 100)]

    @property
    def _default_human_render_camera_configs(self):
        pose = look_at([0.6, 0.7, 0.6], [0.0, 0.0, 0.35])
        return CameraConfig("render_camera", pose, 512, 512, 1,
0.01, 100)

    def _load_agent(self, options: dict):
        super()._load_agent(options, sapien.Pose(p=[-0.615, 0, 0]))

    def _load_scene(self, options: dict):
        self.table_scene = TableSceneBuilder(
            env=self, robot_init_qpos_noise=self.
robot_init_qpos_noise
        )
        self.table_scene.build()

        # create cube
        self.obj = actors.build_cube(
            self.scene,
            half_size=self.cube_half_size,
            color=np.array([12, 42, 160, 255]) / 255,
            name="cube",
            body_type="dynamic",
            initial_pose=sapien.Pose(p=[0, 0, self.cube_half_size]),
        )

        # create target
        self.goal_region = actors.build_red_white_target(
            self.scene,
            radius=self.goal_radius,
            thickness=1e-5,
            name="goal_region",
            add_collision=False,
            body_type="kinematic",
        )

    def _initialize_episode(self, env_idx: torch.Tensor, options:
dict):
        with torch.device(self.device):
```

```
            b = len(env_idx)
            self.table_scene.initialize(env_idx)
            xyz = torch.zeros((b, 3))
            xyz[..., :2] = torch.rand((b, 2)) * 0.2 - 0.1
            xyz[..., 2] = self.cube_half_size
            q = [1, 0, 0, 0]

            obj_pose = Pose.create_from_pq(p=xyz, q=q)
            self.obj.set_pose(obj_pose)

            target_region_xyz = xyz - torch.tensor([0.1 + self.
goal_radius, 0, 0])

            target_region_xyz[..., 2] = 1e-3
            self.goal_region.set_pose(
                Pose.create_from_pq(
                    p=target_region_xyz,
                    q=euler2quat(0, np.pi / 2, 0),
                )
            )
            self.object_list = {"cube": self.obj,
                                "goal": self.goal_region}
    def evaluate(self):
        is_obj_placed = (
            torch.linalg.norm(
                self.obj.pose.p[..., :2] - self.goal_region.pose.p
[..., :2], axis=1
            )
            < self.goal_radius
        )

        return {
            "success": is_obj_placed,
        }

    def _get_obs_extra(self, info: Dict):
        obs = dict(
            tcp_pose=self.agent.tcp.pose.raw_pose,
            goal_pos=self.goal_region.pose.p,
        )
        if self._obs_mode in ["state", "state_dict"]:
            obs.update(
                obj_pose=self.obj.pose.raw_pose,
            )
        return obs

    def compute_dense_reward(self, obs: Any, action: Array, info:
Dict):
        # grippers should close and pull from behind the cube, not
grip it
        # distance to backside of cube (+ 2*0.005) sufficiently
encourages this
        tcp_pull_pos = self.obj.pose.p + torch.tensor(
            [self.cube_half_size + 2 * 0.005, 0, 0], device=self.
device
        )
        tcp_to_pull_pose = tcp_pull_pos - self.agent.tcp.pose.p
        tcp_to_pull_pose_dist = torch.linalg.norm(tcp_to_pull_pose,
 axis=1)
        reaching_reward = 1 - torch.tanh(5 * tcp_to_pull_pose_dist)
        reward = reaching_reward

        reached = tcp_to_pull_pose_dist < 0.01
```

```
        obj_to_goal_dist = torch.linalg.norm(
            self.obj.pose.p[..., :2] - self.goal_region.pose.p[...,
 :2], axis=1
        )
        place_reward = 1 - torch.tanh(5 * obj_to_goal_dist)
        reward += place_reward * reached

        reward[info["success"]] = 3
        return reward

    def compute_normalized_dense_reward(self, obs: Any, action:
Array, info: Dict):
        max_reward = 3.0
        return self.compute_dense_reward(obs=obs, action=action,
info=info) / max_reward

    def get_fitness_score(self):
        # get the fitness score of the current episode
        # returns a tensore of shape (batch_size, )
        # currently, the fitness score is the distance between the
cube and the goal
        # fitness score always the higher the better so we return
the negative distance
        return -torch.linalg.norm(
            self.obj.pose.p[..., :2] - self.goal_region.pose.p[...,
 :2], axis=1
        )
```

## Previous Implementations:
Previous reward function:
```python
def compute_dense_reward(self, obs, action, info):
    reward = torch.zeros_like(info["task_completed"], dtype=torch.
float32)
    # Simple reward for moving toward object
    target_pos = self.object.pose.p
    gripper_pos = self.agent.tcp.pose.p
    distance = torch.norm(gripper_pos - target_pos, dim=-1)
    reward -= distance * 0.1
    return reward
```
Your reward function code has been analyzed, the feedback is as
follows:

## Human Feedback:
The robot moves toward the object but doesn't attempt to grasp it.

## Output:
```python
def compute_dense_reward(self: BaseEnv, obs: Any, action: torch.
Tensor, info: Dict[str, torch.Tensor]) -> torch.Tensor:
    # Initialize reward tensor for batch processing
    reward = torch.zeros_like(info["task_completed"], dtype=torch.
float32)

    target_pos = self.object.pose.p
    gripper_pos = self.agent.tcp.pose.p
    distance = torch.norm(gripper_pos - target_pos, dim=-1)
    reward -= distance * 0.2  # Increased weight for faster
approach
```

```
    is_grasping = self.agent.is_grasping(self.object)
    reward[is_grasping] += 1.0  # Reward for successful grasp

    if "grasped_duration" in info:
        reward += info["grasped_duration"] * 0.05  # Small reward
for maintaining grasp

    return reward
```

# New Chat Session

## Environment code:
```python
{environment_code}
```

## Previous Implementations:
Previous reward function:
```python
{compute_dense_reward}
```

Your reward function code has been analyzed, the feedback is as
follows:
## Human Feedback:
{human_feedback}

Your output should be formatted as a Python code string: "```python
... ```"
- The code should contain the complete `compute_dense_reward`
function

## Output:
```

## C.6 ERROR CORRECTION PROMPTS

Short Horizon Error Correction Prompt

```
You are being given code that is a training environment for a
reinforcement learning-based robot. Code details:
1. `_load_scene`, `_initialize_episode`, and all the methods called
by them, set up the scene.
2. `evaluate` calculates information about the current environment
state in a dictionary. Its output dictionary is given to `
compute_dense_reward` in the parameter `info`.
3. `compute_dense_reward` outputs a final reward.

The code is as follows:
```python
{generated_env_code}
```

This code has an error in it that you need to fix. The stack trace
is as follows:
```python
{error_trace}
```
```

```
The line that threw the error is `{error_line}`.

Please fix it.
1. You cannot take debugging steps, you have to fix it without more
 info.
2. You should only edit `evaluate` and `compute_dense_reward` and
any of their helper methods. Enclose them in markdown ```python <
your code> ``` delimiters.
3. Don't use "```python ... ```" delimiters anywhere else in your
response. You can use `  ` if needed.
3. If you would absolutely have to edit other methods to make the
code work, you should instead say "this requires edits to
restricted parts of the code. I can't continue."
4. Follow the documentation below. Do not violate it.
5. OUTPUT THE WHOLE REVISED FUNCTION, NOT JUST A SNIPPET.

{documentation}
```

**Long Horizon Error Correction Prompt**

```
You are being given code that is a training environment for a
reinforcement learning-based robot. Code details:
1. `_load_scene`, `_initialize_episode`, and all the methods called
by them, set up the scene.
2. `evaluate` calculates information about the current environment
state in a dictionary. Its output dictionary is given to `
skill_reward` in the parameters `prev_info` and `cur_info` – each
represents `evaluate` called on a different environment state.
3. `prev_info` and `cur_info` go through a conversion that turns
their `torch.Tensor` values with batch dimensions into `numpy.
ndarray` values WITHOUT a batch dimension.
3. `skill_reward` outputs a final reward.

The code is as follows:
```python
{generated_env_code}
```

This code has an error in it that you need to fix. The stack trace
is as follows:
```python
{error_trace}
```

The line that threw the error is {error_line}.

Please fix it.
1. You cannot take debugging steps, you have to fix it without more
 info.
2. You should only edit `evaluate` and `skill_reward` and any of
their helper methods.
3. If you would absolutely have to edit other methods to make the
code work, you should instead say "this requires edits to
restricted parts of the code. I can't continue."
4. OUTPUT THE WHOLE REVISED FUNCTION, NOT JUST A SNIPPET.
```

# D HUMAN SURVEYS

## D.1 HUMAN ANNOTATORS

We recruited a total of five preference/evaluation annotators through Upwork Upwork (n.d.). Each chain of preference was generated by one annotator: they would watch an initial video and give a preference, evaluate the result and issue a new preference, and so on until chain termination. Videos were 2-6 seconds in length, evaluation and preference forms took 1-3 minutes to fill out. Annotators were asked to respond within 6 business hours and given a maximum of 24 hours, so one four-step chain would take 2-4 days. Annotators were compensated with $20 each time they provided eight new pieces of data, whether evaluations or preferences.

Expert evaluators were recruited for voluntary effort.

## D.2 SURVEY CONTENT

### D.2.1 PREFERENCE/ALIGNMENT EVALUATION SURVEYS

We conduct three types of surveys:

1. Robot Video Preference, as shown in Survey 1: This survey collects human preference on robot trajectory videos.

2. Video Quality Evaluation, as shown in Survey 2: This survey evaluates how well newly trained robot behaviors align with preference from previous videos.

3. Preference Evaluation, as shown in Survey 3: This survey collects human preferences among videos generated by different methods.

```
  Robot Video Preference

Watch a robot do a task at the video we sent you, then give
feedback.
* Indicates required question

Required Questions:
1. What's your name? Make sure to enter your name the same way
every single time. *

2. Which video are you giving feedback for? Copy-paste the .mp4
file. So if the file is named "video_name.mp4", put "video_name"
here. *

3. Describe the video *

4. Give feedback. Again, be natural! Speak the way you'd want to
speak to your personal robot assistant. Even if the task isn't
inherently interesting, feel free to be creative - ask to change
the order, the location, and more. *

Task Overview:
You're going to watch a robot do a task. Your job is to give the
robot feedback.

Purpose:
- The robot's execution might have problems
- Even if technically fine, it might not meet your preferences
- We want feedback on what you do and don't like
- Goal is to make the robot adapt to you, not vice versa

CRUCIAL GUIDELINES:
1. Procedure:
    - Watch the entire video
```

```
        - Describe the video in the first answer box
        - Enter feedback in the second answer box

    2. Focus on Outcomes:
        - Don't guess how the robot works
        - Don't suggest technical mechanisms
        - Focus on preferences and corrections
        - Describe what you want/don't want

    3. Feedback Limits:
        - Maximum one feedback on overall behaviors
        - Maximum one feedback on object orientation
        - No feedback on speed (technical limitation)

    4. Communication Style:
        - Speak naturally
        - Use everyday language
        - Be specific
        - Context-dependent statements are acceptable

    EXAMPLE SCENARIO:
    Robot sweeping crumbs task

    High Quality Feedback Examples:
    - "The robot didn't finish sweeping. There are crumbs all over the
    place. The crumbs near the couch and TV should have gotten swept up
    ."
    - "I like that the robot brought the crumbs close to the trash can.
     That's a good idea."
    - "Ew, I hate that the robot just left the crumbs near the trash.
    That whole corner of the kitchen is so gross."
    - "I like that the crumbs are in the kitchen, but now it should put
     the broom away."

    Low Quality Feedback Examples:
    - "The robot should work on getting its arm to reach further so it
    can sweep up more crumbs."
    - "This is dumb. It should be calculating more precise trajectories
    ."

    Challenging but Acceptable:
    - "Not that one, the other one." (If context makes meaning clear)

    Note: Your feedback can be longer than examples and include
    multiple ideas. Be thorough and specific.

    This content is neither created nor endorsed by Google.
      Forms
```

---

Video Quality Evaluation

```
Earlier, you watched a video of a robot doing a task and gave
feedback. We tried to incorporate your feedback, and sent you one
or more options in your options-to-choose-from folder.

Instructions:
1. Choose the video you think is best from that folder.
2. For that video only, fill out this form that asks about your
satisfaction with the update.

* Indicates required question
```

```
Basic Information:
1. Copy-paste the name of the video you liked best here. Remove ".
mp4", as you do in the other form. *

Feedback Implementation Assessment:
2. In the most recent video, did the robot incorporate your
feedback that **wasn't met in the previous video**? *
*Mark only one oval.*
o No - the robot didn't incorporate any part of my feedback
o Some - the robot incorporated some parts of my feedback, but not
all of them.
o Yes - the robot incorporated all parts of my feedback.

3. In the most recent video, to what degree did the robot
incorporate your feedback that **wasn't met in the previous video
**? *
*Mark only one oval.*
o None - the robot didn't incorporate my feedback
o Medium - the robot incorporated my feedback to a moderate degree
but not fully.
o High - the robot incorporated my feedback fully.

4. In the most recent video, did the robot get worse with respect
to any of your feedback **compared to the previous video**? *
*Mark only one oval.*
o No - it did not get worse with respect to any part of my feedback.

o Some - it got worse with respect to some parts of my feedback,
not others.
o Yes - it got worse with respect to all parts of my feedback.

5. In the most recent video, did the robot change its performance
in ways you didn't ask for **compared to the previous video**? *
Specifically, change even if not directly/purely negating your
feedback?
For example, if you asked the robot to be neater and smoother but
didn't say anything about speed, and the robot became noticeably
faster in this video compared to the last one, you would say yes.
Else, no.
*Mark only one oval.*
o Yes  *Skip to question 7*
o No   *Skip to question 11*

Progress Assessment:
6. How much **progress** do you feel the robot made on the goal you
 wanted? *
If it got halfway toward the goal your feedback was asking for,
that would be a 5.
*Mark only one oval.*
1 2 3 4 5 6 7 8 9 10
Did the goal fully o o o o o o o o o o The robot did something you
didn't ask for

Unasked Changes:
7. Which of these statements best describes your feelings about the
 change(s) you didn't ask for? *
*Mark only one oval.*
o I strongly dislike them.
o I somewhat dislike them.
o I feel neutrally about them.
o I somewhat like them.
o I strongly like them.
```

```
8. Was the robot doing these same things you didn't ask for in the
previous video? *
*Mark only one oval.*
o Yes
o No

9. In the most recent video, does the robot's performance directly
contradict your feedback? *
*Mark only one oval.*
o Yes
o No

10. Describe the changes you saw that you felt you didn't ask for,
but weren't directly/purely against your feedback. *

Satisfaction Assessment:
11. Are you more satisfied with the robot in the most recent video
than you were in the previous video? *
*Mark only one oval.*
o Yes  *Skip to question 12*
o No   *Skip to question 13*

Satisfaction Level:
12. How much more satisfied are you? *
*Mark only one oval.*
1 2 3 4 5
Not at all o o o o o Completely satisfied

13. How much more dissatisfied are you? *
*Mark only one oval.*
1 2 3 4 5
Not at all o o o o o Completely dissatisfied

Overall Satisfaction:
14. How satisfied are you with the task overall, regardless of your
 feedback? *
*Mark only one oval.*
1 2 3 4 5
o o o o o Very satisfied

This content is neither created nor endorsed by Google.
  Forms
```

Preference Evaluation

```
1. Folder Name *
Please copy and paste the name of the folder.
(e.g., labtest-PlaceSphere2BinWideb44eadad34-ee940ce50d773948).

2. Please rank the videos from 1 to N based on your preference,
with 1 being the best.
For example, if there are six videos, a rank of 3,4,2,1,5,6
indicates you love 4.mp4 the best.
                1    2    3    4    5    6    7    8    9
Video1          o    o    o    o    o    o    o    o    o
Video2          o    o    o    o    o    o    o    o    o
Video3          o    o    o    o    o    o    o    o    o
Video4          o    o    o    o    o    o    o    o    o
Video5          o    o    o    o    o    o    o    o    o
Video6          o    o    o    o    o    o    o    o    o
```

```
Video7          o   o   o   o   o   o   o   o   o
Video8          o   o   o   o   o   o   o   o   o
Video9          o   o   o   o   o   o   o   o   o
*Mark only one oval per row.*
```

### D.2.2 SEMANTIC MATCH EVALUATION SURVEY CONTENT

Expert evaluation

```
Thank you for doing this expert evaluation for us! Our project aims
 to use LLMs to take natural language feedback and generate dense
reward functions. Our goal with this evaluation is to measure
semantic match between feedback and reward function - does the
generation make sense given the feedback?

Context:
We train Franka arms in several ManiSkill 3 environments - see
video.
report.md has four separate sections:
"Original feedback" - what the annotator said.
"Grounded feedback" - a generated version of the feedback that
attempts to ground it in environment + prior performance.
"Staging plan" - a generated plan for writing a staged reward,
similar to a task plan but with some differences.
"Code" - two generated functions, `evaluate` and `
compute_dense_reward`. `compute_dense_reward` is the actual reward
function, `evaluate` is a helper function that packages scene info
and defines success. You can mainly read `compute_dense_reward`, and
 just refer back to `evaluate` as needed (e.g. for success
definition).
We will ask you to evaluate these purely on whether they make sense
 given the feedback - common sense, logic, semantics. You do not
need to worry about trying to predict whether the reward will
actually work or not.

Start by watching the video to get some context!

NOTE: in the bin task, the bins cannot be moved. In the push ball
task, the target cannot be moved. The robot's starting position
cannot be changed in either case.
If a feedback asks for that, it's okay if it's in the stage plan,
but it should not be in the code.

To what degree does the grounded feedback correctly interpret the
original feedback?
[Likert scale 1-7]

Note: please evaluate the staging as it relates to the grounded
feedback, not the original feedback. We want to separate evaluation
 of each step.
To what degree does the staged reward plan seem like an accurate
and precise attempt at incentivizing the desired behavior?
[Likert scale 1-7]

Are there missing steps?
Yes/No

Are there unnecessary steps?
Yes/No
```

```
Is the strategy bad overall?
Yes/No

If you'd like, describe the problems in a sentence or two.

Note: please evaluate the coding as though the staging were
reasonable, whether or not it actually is - again, trying to
evaluate each part separately.
To what degree does the reward code seem like an accurate and
complete attempt at incentivizing the desired behavior?
[Likert scale 1-7]

Does the reward function miss part or all of the stage plan?
Yes/No

Is the reward function so sparse that it's not even semantically
aligned? Note that somewhat sparse rewards can be semantically
aligned, even if they won't be effective for training. This is
asking for reward so sparse that it's not, or only trivially,
semantically aligned.
Yes/No

At a semantic level, are there logic errors? Contradictions,
circular dependencies, etc.
Yes/No

Are there common sense errors?
Yes/No

If you'd like, describe the problems in a sentence or two.

Even if all the generations technically seem correct given the
feedback, there's obviously nuance in language that may be missed.
Would you have interpreted and approached this similarly?
Yes/No

If you answered anything less than 7 for any of the overall quality
 questions, to what degree do you think this was due to difficult
human feedback?
[Likert scale 1-7]
```

### D.3  TAXONOMY

Taxonomy definitions

- **3rd-person vs. 2nd-person vs. no-pov**: how the preference addresses the embodied agent
- **verbose**: concepts explained in lots of words, more than may be necessary
- **colloquial**: concepts explained colloquially, casually, or with imprecise (but not ambiguous) terms
- **context-dependent**: requires info from the demonstration or previous preferences to be understood; environment code is not sufficient
- **ambiguous**: even context is not enough, and the preference is up to interpretation (rare)
- **multi-part**: multiple concepts within one preference
- **directive vs. suggestive vs. curious**: preference tone. *Directive* is command-like, *suggestive* is phrased as a question but clearly instructional, and *curious* is a genuine question about the agent's abilities or potential
- **distraction**: contains distracting language that may be incorrectly incorporated into the reward function

- **contradiction**: preference that directly contradicts prior preference
- **preferential**: adds to or changes task requirements
- **corrective**: doesn't add to or change the task requirements, instead asks for improvement on current requirements
- **preferential**: adds to or changes the task requirements
- **goal-related**: affects the definition of success
- **behavioral**: affects other parts of the reward landscape
- **inadvertently long-horizon**: creates an impossible masking problem in the short-horizon continuous control setting
- **affirming**: positive about some aspect of the task that should be propagated
- **asking for change**: asking for some aspect to be changed
- **physically difficult**: difficult for the test agents, e.g. throwing a ball
- **physically impossible**: impossible for the test agents, e.g. picking up a ball that is much wider than the gripper.
- **multi-objective**: preference containing multiple complex goals that need to be achieved simultaneously. Restricted (and rare)
- **specific position**: complex or specific natural language description of a position
- **specific orientation**: complex or specific natural language description of an orientation
- **illegal**: requires edits to code outside of `compute_dense_reward` and `evaluate`.

## E  DETAILED METRICS

- **Alignment**
    - **Percent more satisfied (MS)**: given a binary question asking whether or not they were more satisfied with the current video than with the previous video given their preference, for what percentage of preferences did the annotator say were more satisfied?
    - **Satisfaction score (SS)**: annotators were given a five-point Likert scale to indicate their degree of (dis)satisfaction, based on their answer to the binary satisfied-yes-no question. In both cases, 1 was the smallest amount of (dis)satisfaction, and 5 was the highest amount. Satisfaction score is calculated by shifting both from 1-5 to 0-4 as 1 means neutrality, negating the Likert score value in dissatisfaction cases, and rescaling from 0 to 10.
    - **Percent chosen (PC)**: when directly comparing policy rollouts on a single preference, frequency of the given method's rollout being selected over ROSETTA's
- **Optimizability**
    - **Success rate (SR)**: success rate of best policy across all variants for a given preference
    - **Percent success >50**: percent preferences where best policy has success rate higher than 50%
- **Semantic match**
    - **Grounding**: normalized 7-point Likert scale score on semantic match between original and grounded preference
    - **Staging**: normalized 7-point Likert scale score on semantic match between grounded preference and staging plan
    - **Coding**: normalized 7-point Likert scale score on semantic match between staging plan and reward code
    - **Cascading**: product of Grounding, Staging, and Coding. Generally evaluated only for preferences expert evaluators consider reasonable, given quickly compounding errors.

| Hyperparameter | Value |
|---|---|
| Learning rate | 3e-4 |
| Discount factor ($\gamma$) | 0.8 |
| GAE parameter ($\lambda$) | 0.9 |
| Number of minibatches | 32 |
| PPO epochs | 8 |
| Value function coefficient | 0.5 |
| Entropy coefficient | 0.0 |
| Gradient norm clipping | 0.5 |
| Target KL divergence | 0.1 |
| Actor Size | 256*2 |
| Critic Size | 256*2 |

Table 3: PPO Hyperparameters

| Hyperparameter | Value |
|---|---|
| Learning rate | 3e-3 |
| Discount factor ($\gamma$) | 0.2 |
| Update Coefficient ($\tau$) | 0.5 |
| # of Gradient Step | 8 |
| Batch Size | 1024 |
| Policy Model Size | 512*2 |

Table 4: SAC Hyperparameters

| Environment Name | Total Timesteps | Episode Length | Description |
|---|---|---|---|
| Pick1Cube | 10M | 50 | Cube picking |
| Pull1Cube | 10M | 50 | Cube pulling |
| PushBall | 20M | 50 | Ball pushing |
| PlaceSphere2BinWide | 100M | 200 | Sphere placement in 2 bins |

Table 5: Short-horizon Environment Setups

# F    REINFORCEMENT LEARNING DETAILS

## F.1    SHORT-HORIZON SETUP

We implement our approach using PPO for training agents on four short-horizon environments. The implementation is based on the ManiSkill environment, which provides a realistic physics-based simulation platform for robotic manipulation. The policy is trained using parallel environments (num_envs = 1024) to improve sampling efficiency and training stability. We use state-based observation and 7-dimensional pd_joint_delta action.

The training hyperparameters are given in Table 3. The training steps of different environments are given in Table 5.

## F.2    LONG-HORIZON TRAINING SETUP

**RL Problem Setup**    We model the robot manipulation task as a Markov decision process denoted by the set ($\mathcal{S}$, $\mathcal{A}$, $\mathcal{P}$, $\mathcal{R}$, $\gamma$), representing the state space, the action space, the transition function, the reward function, and the discount factor. A policy $\pi$ is a mapping from the observation state space $\mathcal{S}$ to a probability distribution over the robot action space $\mathcal{A}$.

The observable state space $\mathcal{S}$ consists of the position of the object in the scene, the gripper's position, the interaction between the robot and the object (e.g., whether the robot is picking up an object), and the current stage of task progression. The observation space for each long-horizon environment is provided in Table 6.

For the action space $\mathcal{A}$, we use a subset of parameterized primitive skills from MAPLE (Nasiriany et al. (2022a)), specifically: Pick, Place, and Push. The descriptions of these skills are provided in Table 7. The action space $\mathcal{A}$ is 9-dimensional, where the first three values represent the probability distribution over the available skills, and the remaining six values serve as parameters for the selected skill. Each primitive skill execution typically completes within 100 low-level actions. If execution exceeds 250 low-level actions, the robot stops and retracts to its home position.

| Environment | Observation Space |
|---|---|
| ThreeCubes | 3D positions of the red, green, and purple cubes; grasp status of each cube; robot's gripper position; and the current stage of the task. |
| ObjectsAndDrawer | Positions of the drawer, drawer handle, apple, and soup can; grasp status of the apple and soup can; robot's gripper position; and the current stage of the task. |
| ObjectsAndBins | 3D positions and grasp status of the tennis ball, baseball, orange, and apple; positions of two bins; robot's gripper position; and the current stage of the task. |
| SphereAndBins | 3D positions and grasp status of the sphere; positions of two bins; robot's gripper position. |
| BallAndTarget | 3D positions and grasp status of the ball; positions of the target; robot's gripper position. |

Table 6: Observation Space for Long Horizon Environments

| Skill | Description |
|---|---|
| Pick | Moves the end-effector to a specified (x, y, z) location and closes the gripper to grasp an object. |
| Place | Moves the end-effector to a specified (x, y, z) location and opens the gripper to release an object. |
| Push | Moves the end-effector to a specified (x, y, z) location, then applies a displacement $(\delta x, \delta y, \delta z)$ to push an object. |

Table 7: Parameterized Primitive Skills

The task horizon is inferred from the generated reward function and varies depending on human feedback. The remaining RL setup is consistent with the short-horizon case.

**Training Setup**  We train a hierarchical policy (Nasiriany et al. (2022a)) with two 512-dimensional layers using Soft Actor-Critic (SAC) (Haarnoja et al. (2018b)). Hyper-params are given in Table 4.

Training terminates early when evaluation accuracy reaches 0.9 or higher. If this threshold is not met, training continues until 200 million lower-level steps have been completed. The same training configuration is applied across all three long-horizon environments for every generated reward function, with hyperparameters optimized to minimize training time.

To further improve sample efficiency and reduce training time, we apply a balanced rollout buffer that ensures rollouts are evenly sampled across all task stages, leading to faster learning, especially for the robot to learn later stages.

## G  COST ANALYSIS

Here, we compare cost of ROSETTA to Eureka and Text2Reward. We note that Eureka and Text2Reward both use all rounds of reward iteration to optimize performance on a single task, whereas ROSETTA adds more overhead in a single round so that it can adapt in one step to a new task. The strategies that require these extra tokens, particularly staging and full code rewrite at each step, allow ROSETTA to outperform on task-changing preferences.

Terms:

- Terms:

- – `c_code`: average cost of writing one reward function with `o1-mini`
- – `c_lang`: average cost of grounding and staging with `gpt-4o` (significantly less than `c_code`)
- – `c_train`: robot training cost
- – `num_iterations`: number of iterations that the entire method is run for one task (ROSETTA: 1, Text2Reward: 3, Eureka: 5)
- – `c_reflection`: average cost of doing one reward reflection

- Expressions:
  - – ROSETTA: $7 * (\texttt{c\_lang} + \texttt{c\_code}) + 3 * \texttt{c\_train}$ - 7 variants generated due to verification questions, 3 variants trained and shown to human
  - – Text2Reward: $3 * \texttt{c\_code} + 3 * \texttt{c\_train}$ - Text2Reward experiments use 3 iterations per task
  - – Eureka: $5 * \texttt{c\_reflection} + 400 * \texttt{c\_code} + 400 * \texttt{c\_train}$ - Eureka experiments use 5 iterations per task, each containing an evolutionary search with 5 evolutions and 16 branches per evolution.

We note that Eureka and Text2Reward experiments use less powerful LLMs than ROSETTA experiments do; in reality, it is likely that all three methods will be closer in cost than they seem here, because more powerful LLMs raise $\texttt{c\_*}$ costs but reduce hyperparameters like number of evolutions and evolution branch batch size. However, we expect the ranking to stay as-is.

For training costs, short-horizon tasks take 20-90 minutes on on etitan rtx or rtx 2080; long-horizon tasks take 8-12 hours on the same, though they can be sped up on higher-end GPUs.

