# OpenReview forum: "ROSETTA: Constructing Code-Based Reward from Unconstrained Language Preference"
_ICLR.cc/2026/Conference — ICLR 2026 Poster_

### Official Review · Reviewer_9E3U · 2025-10-29

**Soundness:** 3
**Presentation:** 3
**Contribution:** 2
**Rating:** 6
**Confidence:** 4

**Summary:**

This paper introduces the ROSETTA framework, which uses LLMs to directly translate human natural language preferences into executable reward code. It addresses a problem in embodied intelligence: enabling agents to adapt to human preferences in real-time and online. ROSETTA can generate new reward functions and train policies, demonstrating performance significantly superior to existing baselines.

**Strengths:**

- The paper addresses a critical problem by directly modeling human language, tackling the challenge of aligning with unconstrained feedback.

- The capability for online adaptation is a major strength. The agent can adapt to a completely new human preference within a single interaction, which enables dynamic and continuous interactive alignment.

- The evaluation methodology is comprehensive. The authors propose three-dimensional evaluation metrics that assess the policy from multiple perspectives, establishing a strong benchmark for this problem.

- The framework was evaluated on a large-scale set of human preferences, achieving very high success rates and human satisfaction.

**Weaknesses:**

- Portability is questionable: The method relies heavily on domain knowledge. Migrating it to a new robot or a new environment would likely require significant manual setup.

- The policy learning process is simplified: While the paper highlights "single-step adaptation," it glosses over the policy learning process. The overall training loop is complex: after the reward function is generated, a policy must be fully trained in the simulator before the user can watch it and provide new feedback. The authors should provide details on the time and resources required for this training phase, as well as the iteration time for the LLM to generate the reward function.

- The necessity of the human selection step is unclear: The system relies on training multiple reward code variants and then requires a human evaluator to watch videos and select the best one. It is unclear if this human-in-the-loop selection step is essential and how performance would be affected without it.

- Experiments are simulation-only: Despite the large number of preferences and comprehensive experiments, the evaluations are conducted exclusively in simulation. Unlike a paper proposing a purely novel algorithm, this paper's primary contribution is a feasible framework. Therefore, it needs to demonstrate the feasibility of the full real-sim-real loop, perhaps with a simple case study. Simulation-only experiments overlook numerous potential errors that can occur in this cascaded process.

**Questions:**

1.Has the author verified the entire process in a real robot scenario?

---

> ### Author Response · Authors · 2025-11-29
> **Thank you for your thoughtful review!**
>
> Thank you so much for your kind words and thoughtful review! We are glad to hear that you feel that this is a “critical problem”, that “the capability for online adaptation is a major strength”, etc. We’re excited that it resonated with you. We’ll now discuss your concerns and look forward to your feedback!
>
> > Portability is questionable: The method relies heavily on domain knowledge. Migrating it to a new robot or a new environment would likely require significant manual setup.
>
> This is a very important point. That said, the domain knowledge is less about ManiSkill and more about general physical principles, e.g. considering all three dimensions when setting a target position. There is some ManiSkill-specific code, but it is also primarily related to e.g. boolean logic or geometry, and could be translated easily. We would be happy to try porting to another simulator for the camera-ready, if that would be useful.
>
> > The policy learning process is simplified: While the paper highlights "single-step adaptation," it glosses over the policy learning process. The overall training loop is complex: after the reward function is generated, a policy must be fully trained in the simulator before the user can watch it and provide new feedback. The authors should provide details on the time and resources required for this training phase, as well as the iteration time for the LLM to generate the reward function.
>
> Great question! The hyperparameters are kept constant as they are not the focus of the paper, and are available in appendix tables 3 and 4. The training costs are relatively low: short-horizon tasks take 20-90 minutes on one titan rtx or rtx 2080; long-horizon tasks take 8-12 hours on the same, though can be sped up on higher-end GPUs. We have added these to section G. Generation costs are available in section G as well. Let us know if this is helpful!
>
> > The necessity of the human selection step is unclear: The system relies on training multiple reward code variants and then requires a human evaluator to watch videos and select the best one. It is unclear if this human-in-the-loop selection step is essential and how performance would be affected without it.
>
> This is a great question and important to discuss! This relates to our response to reviewer Tpkx: the human annotations are not part of the method; they are the end goal. We do not know the success condition or fitness function, and the human annotation is the only source of truth. Furthermore, the success condition keeps changing as the human's preference keeps changing: \algo is single shot, and multiple iterations of \algo aren't meant to improve performance. They occur only when a new, potentially task-changing, preference is issued.
>
> Let us know if this is helpful! We are very happy to discuss further, and we have emphasized this point in Section 3: Problem Formulation. Thank you for the clarification - we feel that the paper benefits from it.

---

### Official Review · Reviewer_zcpx · 2025-10-31

**Soundness:** 3
**Presentation:** 2
**Contribution:** 3
**Rating:** 6
**Confidence:** 3

**Summary:**

This paper proposes Rosetta, a framework that leverages foundation models to ground and disambiguate natural language preferences, construct multi-stage reward functions, and implement them via code generation. The core contributions include the Rosetta framework itself, an evaluation framework measuring alignment, semantic match, and optimizability, and extensive experimental validation on 116 human preferences across five manipulation tasks. Key results report an 87% average success rate, 86% human satisfaction, and outperformance over state-of-the-art baselines like Eureka and Text2Reward, particularly for non-corrective preference types.

**Strengths:**

**Well-Structured Framework**: The three-module design (Preference Grounding, Staging, Coding) effectively handles unconstrained language by disambiguating references and structuring rewards, as illustrated in Fig. 2 and described in Sec. 4. Integration of domain knowledge through verification questions in the Coding module (Sec. 4.3) enhances reward reliability, with 95.38% of generated functions running after error correction. The framework supports online adaptation to dynamic preferences, maintaining performance over multiple iterations (Tab. 1), which is a significant advancement over prior fixed-goal methods.

**Comprehensive Evaluation**: The proposed evaluation framework (Sec. 3.1) with alignment, semantic match, and optimizability metrics provides a holistic assessment beyond standard success rates, addressing the subjectivity of human preferences. Extensive human evaluation on 116 preferences across five environments (Sec. 5.1) demonstrates robustness, with results broken down by preference type (Fig. 3, Fig. 4) and language/content diversity (Fig. 5, Fig. 7). High performance metrics (86% human satisfaction, 87% success rate) across short-horizon and long-horizon tasks indicate practical utility for real-world applications.

**Weaknesses:**

**Insufficient Details and Reproducibility**: Hyperparameters for policy training are not detailed in Sec. 4.4 or Sec. 5.1, making replication difficult. Computational costs and resource requirements are omitted, despite the use of multiple LLM calls per preference in Sec. 4. The policy selection process in Sec. 4.4 relies on human annotators choosing from variants, but no inter-annotator agreement or selection criteria are reported, potentially introducing bias.

**Limited Generalization**: The framework is task-agnostic but tested only in ManiSkill 3 environments; transfer to other simulators or domains is not evaluated, as acknowledged in Sec. 6 (Limitations). Semantic match evaluation in Sec. 3.1 relies on expert assessment, which may not scale and lacks objectivity compared to automated metrics; no inter-rater reliability is reported. The approach depends on specific foundation models (gpt-4o, ol-mini), and while open-source model tests in Sec. 5.2.1 show promise, full ablation on model choices is missing.

**Questions:**

1.	How does ROSETTA perform with long sequences of preferences in terms of performance degradation and computational efficiency? Does the foundation model context window become a bottleneck? (Sec. 5.2.3; Table 1)
2.	What is the impact of alternative error correction strategies (beyond iterative ol-mini) on success rate and human satisfaction? Could reinforcement learning for code correction be beneficial? (Sec. 4.3; Sec. 5.2.1)
3.	How generalizable is ROSETTA to non-manipulation embodied tasks? Are there module design limitations requiring modification? (Sec. 1; Sec. 5.1)

---

> ### Author Response · Authors · 2025-11-29
> **Thank you for your thoughtful review!**
>
> Thank you so much for your kind words and thorough review. We are very encouraged to hear that you consider the framework well-structured and the evaluation comprehensive. We will now discuss your critiques and questions, and hope to resolve as much as possible.
>
> > Insufficient Details and Reproducibility: Hyperparameters for policy training are not detailed in Sec. 4.4 or Sec. 5.1, making replication difficult. Computational costs and resource requirements are omitted, despite the use of multiple LLM calls per preference in Sec. 4.
>
> Thank you for highlighting these important details!
> - Hyperparameters: hyperparameters are available in Tables 3 and 4 in the appendix! Because the paper emphasis is on the reward functions, not the learning algorithm, we kept these to the appendix - let us know you feel this should change.
> - Generation costs can be found in section G. For training costs, short-horizon tasks take 20-90 minutes on one titan rtx or rtx 2080; long-horizon tasks take 8-12 hours on the same, though can be sped up on higher-end GPUs. We have added these to section G. Let us know if this is helpful!
>
> > The policy selection process in Sec. 4.4 relies on human annotators choosing from variants, but no inter-annotator agreement or selection criteria are reported, potentially introducing bias.
>
> This is a great point to discuss. We structure the selection this way because the goal is to adapt to individual preferences. There is no fixed goal - instead, whatever the person who issued the preference wants, is the target. The selection criteria is simply whichever one they like best because of this motivation. We have emphasized this in section 3: problem formulation.
>
> That said, we do very much agree that while the preference-giver’s opinion is the ground truth for alignment/preference adaptation, having other raters audit the match would be a useful objective signal about general human alignment and/or semantic match. We would be happy to include this going forward.
>
> > Limited Generalization: The framework is task-agnostic but tested only in ManiSkill 3 environments; transfer to other simulators or domains is not evaluated, as acknowledged in Sec. 6 (Limitations).
>
> We agree, and to some degree consider this to be an inherent limitation of code-based representations of environment and reward. To offset this:
> - The benefits of code-based reward are significant and hopefully outweigh these
> - ROSETTA is highly task-agnostic, making it more general than the baselines
> - The domain knowledge given does involve ManiSkill-specific code, but is primarily text or easily translatable mathematical code. This may make the transfer more straightforward.
> We hope these help mitigate the concern of lack of generality across simulators! Let us know if you feel this warrants an addition to the paper.
>
> > Semantic match evaluation in Sec. 3.1 relies on expert assessment, which may not scale and lacks objectivity compared to automated metrics; no inter-rater reliability is reported.
>
> We very much agree with these critiques of the semantic match evaluation. Because the generations are long-form and involved, this ended up mainly being a matter of capacity - the labor we needed was scarce and expensive. Going forward, we would very much like to incorporate your suggestions: both more robust expert evaluation, and an objective metric e.g. using embeddings. For now, we aim to mitigate this with the quality of the raters, who are field experts.
>
> > The approach depends on specific foundation models (gpt-4o, ol-mini), and while open-source model tests in Sec. 5.2.1 show promise, full ablation on model choices is missing.
>
> This is another important and cogent critique. As before, the primary concern is with cost and capacity; next steps would definitely include more models, especially as the landscape has evolved. We would be very glad to hear your suggestions for model types as well!

---

### Official Review · Reviewer_Tpkx · 2025-11-02

**Soundness:** 3
**Presentation:** 3
**Contribution:** 3
**Rating:** 4
**Confidence:** 3

**Summary:**

The paper tackles an important problem of adapting the behavior of embodied systems to natural language feedback. It adopts the approach of using foundational models to understand diverse and free form language commands and feedback to generate code that steers the actions of the agent towards the desired objectives. Further, the authors present additional evaluation metrics to analyse the behavior of the method beyond success rate, in terms of human satisfaction, feasibility and semantic similarity of the adapted behavior. The authors demonstrate that they out-perform the baselines across multiple tasks and users, with improvements in human satisfaction.

**Strengths:**

- The papers focus on the setting of unconstrained human preferences where the task is unknown initially and the agent has to adapt on the fly. It is a very important skill that is required for AI systems to be deployed in the real-world.
- The paper is very well written with very descriptive figures, with the motivation, method and experiments clearly laid out.
- The authors transfer their method across both simulation and real-world tasks which shows the strong applicability of the proposed solution.
- The paper includes extensive qualitative examples from their experiments to show the different success and failure cases, while also thoroughly discussing the limitations of the paper.

**Weaknesses:**

- The method and the different components are explained well, but is it difficult to understand the individual contributions of each module to the final method? It would be interesting to see other additional baselines that run an LLM end2end for the whole tasks (with prompts to reason or use CoT to perform staging, ground and code generation by the same model)? This could make the contribution of having this modular stack as compared to making a single call to the LLM.
- The authors include average reward values across different models in their method, but what is the average reward achieved by the different baselines?
- In this method, it requires humans in the loop to provide the fitness function, which seems to be a very expensive process. It would be very interesting to talk about the sample efficiency and human effort required in the overall method?
- Overall, the paper seems very solid, with some missing details in the presentation. I would be happy to raise my score as my questions are answered.

**Questions:**

- Do we expect Eureka to perform similarly if you can provide it with the correct fitness function at each step?  Is it an upper bound on performance? A small explanation of how the baselines were implemented under the interactive setting would be helpful to understand the differences from prior work.
- What instructions are provided to the annotators? Because since the tasks in simulation have a single goal, how does the diverse preferences arise under the same objective for the relatively simpler tasks? Could the authors include some information about the number of participants, and other details about the experiments?
- For the real-world tasks, it would be helpful for reproducibility to understand the implementation details.

---

> ### Author Response · Authors · 2025-11-29
> **Thank you for your thoughtful review!**
>
> Thank you very much for your kind words and thoughtful review! We are so glad to hear that you consider the paper “very well written”, that you consider this a “very important skill”, and that you liked the qualitative examples. We also appreciate your cogent critiques and questions, and will attempt to answer them now.
>
> > The method and the different components are explained well, but is it difficult to understand the individual contributions of each module to the final method? It would be interesting to see other additional baselines that run an LLM end2end for the whole tasks (with prompts to reason or use CoT to perform staging, ground and code generation by the same model)? This could make the contribution of having this modular stack as compared to making a single call to the LLM.
>
> Our ablation section attempts to isolate the value of each module by removing it and showing the drop in performance, particularly on alignment metrics that don't benefit from an optimizable-but-irrelevant reward function. We are actively working on an experiment according to your recommendation, which we feel will further enhance the paper! We will add it to the camera-ready.
>
> > The authors include average reward values across different models in their method, but what is the average reward achieved by the different baselines?
>
> We do not report baseline comparisons using reward because we observed clear reward-hacking behaviors in several baselines, which makes reward an unreliable metric for cross-method comparison. In our setting, humans provide a new instruction to the robot, and a baseline method may generate a reward function that is easy for the agent to optimize but does *not* actually reflect the human instruction. This can lead to artificially high or low reward values that are biased and not meaningful. For this reason, the fairest evaluation is to ask humans to rate the learned policies directly. Optimizability metrics are about optimizability, not correctness.
>
> We use reward only for the internal ablations of our own method, mainly for two reasons. First, ROSETTA separates task grounding from reward design, which substantially reduces the risk of instruction–reward mismatch and makes the reward signal a more faithful indicator of the quality of the generated reward function. Second, we had limited human annotation capacity, so reward-based analysis was necessary for these additional ablations.
>
> > In this method, it requires humans in the loop to provide the fitness function, which seems to be a very expensive process. It would be very interesting to talk about the sample efficiency and human effort required in the overall method?
>
> This is a great question! It’s true that human annotations are expensive and sample inefficient, but the human annotations are not part of the method; they are the end goal. This makes human-provided fitness functions inherent to the pipeline, rather than an expensive process that could be substituted with something cheaper.
>
> Furthermore, in deployment, the preferences would come from the person the agent was assisting. One could analogize our use of human annotators to a focus group testing a product: they are paid, but that payment is not part of the cost of manufacturing the product for the end user.
>
> Let us know if this is helpful! We are very happy to discuss further, and we have emphasized this point in Section 3: Problem Formulation. We think the added clarity will benefit the paper - thank you for the note.
>
> > Do we expect Eureka to perform similarly if you can provide it with the correct fitness function at each step? Is it an upper bound on performance? A small explanation of how the baselines were implemented under the interactive setting would be helpful to understand the differences from prior work.
>
> - Given the correct fitness function at each step, Eureka will likely improve in performance. However, this would require writing code specifying the success condition, and therefore would require ROSETTA’s prompts, which would then make the resulting method a hybrid rather than pure Eureka.
> - Baselines were implemented by using the original prompts and adding ManiSkill documentation for Eureka. In the interactive setting, we gave the new preference to the same prompt. This was to avoid confounding the comparison by changing their prompting/ICL approach and the scope the methods were built for.
> Thank you for calling for clarification - we have made this clear in section 5.2: comparing to baselines.
>
> > For the real-world tasks, it would be helpful for reproducibility to understand the implementation details.
>
> Definitely! We do sim2real transfer and do not notice gap in the tasks we ran. The policy is trained in simulation with the state dictionary observation space. For real world, the observation dictionary is created using FoundationPose. This is reliable and so far has not shown degradation. We have added this explanation to section 5.2.5: Real robot experiments!

---

### Official Review · Reviewer_BmrL · 2025-11-04

**Soundness:** 3
**Presentation:** 2
**Contribution:** 2
**Rating:** 4
**Confidence:** 3

**Summary:**

This paper proposes a multi-stage LLM framework that is used to design reward functions that match indicated preferences at each time-step in open-ended behavior generation tasks. The paper constructs detailed prompts to help with this design process and appears to show improvement in human satisfaction above the baselines.

**Strengths:**

- I really appreciate figure 5, the authors have a very thorough categorization of the different ways that users can express their preferences.
- The paper appears to improve over the results of its baselines.
- The paper provides an interesting, somewhat new goal of generating arbitrary behaviors in each iteration as selected by a human annotator.

**Weaknesses:**

Most of the "weaknesses" in this work are placed in the questions category below. I am uncertain if similar levels of tuning was done on the baselines relative to how much work was done in designing the prompts for Rosetta and would like to understand that better before being able to make a final judgement on the work.

## Very minor (does not affect my score)
- This paper puts me in mind of two papers, "Efficiently Generating Expressive Quadruped Behaviors via Language-Guided Preference Learning" by Clark et al. and "In-Context Preference Learning", but Yu et. al. which also have a similar iterative loop that uses human preferences to select amongst generated candidates. You may want to look at these works as neither are cited but seem highly related.

**Questions:**

- I don't understand the amount of human annotation used in these. The paper points to section F.2 but the relevant content is not there? Where are the details on your experimental protocol for the annotators? Is it one human watching 80 videos? 80 different people? How long are the experiments for them?
- I'm not sure what is meant by Rosetta's "domain knowledge", a frequent claim throughout the paper. Where is the domain knowledge contained in Rosetta as opposed to in the LLMs that comprise Rosetta? More particularly, in what way does Rosetta have domain knowledge that your baselines does not (in particular, I'm responding to line 153)? Do you just mean the prompts in section C? If so, are similar prompts provided to your baselines?
- What LLMs are used for the Eureka and Text2Reward baselines? Are they also using 4o and o1-mini?
- Can you provide more information on the protocol used by the roboticists to evaluate your reward function? The text points to section D in the appendix but section D appears to be about something else.
- In 372 to 377, what are these scores that are being given? Is it the average satisfaction score across several categories? Given that the performance degrades quite a bit when you paraphrase the prompts, why do the authors claim that "the specific prompt matters less than the conceptual structure"?
- Why does the Sat. Score fall off so much after the first round?
- In the real robot experiments, are the "two-iteration preference chains" generated by human annotators? What is the state dictionary that must be populated there?
- How much did this all cost, particularly the annotations? This is relevant information for those who might choose to build on this paper.

---

> ### Author Response · Authors · 2025-11-29
> **Thank you for your thoughtful review!**
>
> Thank you for your kind words about the figures, improvement over baselines, and goal, as well as your detailed and thoughtful review.
>
> > I am uncertain if similar levels of tuning was done on the baselines relative to how much work was done in designing the prompts for Rosetta and would like to understand that better before being able to make a final judgement on the work.
>
> This is a great question! For Eureka, we use its original prompts along with extensive ManiSkill documentation. For Text2Reward, we use its original prompts as they are already built for ManiSkill. We do not alter them beyond simulator documentation because we aim to use them as baselines. We consider ROSETTA’s prompts to be part of its value: the performance discrepancy demonstrates the importance of scaffolding and domain knowledge when adapting to open-ended, real-world scenarios like unconstrained preference adaptation. It is likely that this setting was not considered during Eureka and Text2Rewards’ development.
>
> > This paper puts me in mind of two papers, "Efficiently Generating Expressive Quadruped Behaviors via Language-Guided Preference Learning" by Clark et al. and "In-Context Preference Learning", but Yu et. al. which also have a similar iterative loop that uses human preferences to select amongst generated candidates.
>
> These are very helpful. "Efficiently Generating..." presents another compelling format for rewards from language; we appreciate that it aims to address the setting where the task is unknown, as we aim to. ICPL uses a mechanism like our grounding, where the human feedback is paired with the demonstration directly in the prompt. It makes sense that it works in iteration, and we aim to combine it with other approaches to adapt in a single shot and then to evolving preferences. We have added these to our related works - thank you for pointing us to them.
>
> > I don't understand the amount of human annotation used in these. The paper points to section F.2 but the relevant content is not there? Where are the details on your experimental protocol for the annotators? Is it one human watching 80 videos? 80 different people? How long are the experiments for them?
>
> Thank you for calling out this important detail, and our apologies for the incorrect section! In our current appendix section D, **we will add a subsection D.1 for annotator info** and add a subsection D.3 for the existing survey content. Todo. To answer your questions:
> - A total of five annotators watched videos
> - Each chain of preference was generated by one annotator: they would watch an initial video and give a preference, evaluate the result and issue a new preference, and so on until chain termination.
> - Videos were 2-6 seconds in length, evaluation and preference forms took 1-3 minutes to fill out
> Annotators were asked to respond within 6 business hours and given a maximum of 24 hours, so one four-step chain would take 2-4 days.
>
> > I'm not sure what is meant by Rosetta's "domain knowledge", a frequent claim throughout the paper. Where is the domain knowledge contained in Rosetta as opposed to in the LLMs that comprise Rosetta? More particularly, in what way does Rosetta have domain knowledge that your baselines does not (in particular, I'm responding to line 153)? Do you just mean the prompts in section C? If so, are similar prompts provided to your baselines?
> > What LLMs are used for the Eureka and Text2Reward baselines? Are they also using 4o and o1-mini?
>
> We answer these together as they are both related to informative comparison with baselines!
> - Yes, the domain knowledge is referring to the prompts in section C: i.e. general language guidance on how to approach the problem. We do not provide the prompts to the baselines because we aim to demonstrate the importance of such information to getting successful results on an open-ended and dynamic problem; adding our prompts to Eureka and Text2Reward would confound the comparison.
> - Because none of the three methods are related to actually training language models, we agree the LLM should be held constant. So, as you guessed, we do use 4o and o1-mini for Eureka and Text2Reward; the original papers used GPT-4. Even in this setting, ROSETTA outperforms on task-changing preferences.
>
> Thank you for ensuring this detail is clarified: we have added and emphasized in Sec. 5.1 Baselines.

---

> > ### Author Response · Authors · 2025-11-29
> > **Continuation**
> >
> > > Can you provide more information on the protocol used by the roboticists to evaluate your reward function? The text points to section D in the appendix but section D appears to be about something else.
> >
> > Yes, apologies for the omission and thank you for your very careful read! We have added that survey to section D.2. Please feel free to check it out: we asked a series of binary (e.g. “are there missing steps?”) and Likert (“To what degree does the reward code seem like an accurate and complete attempt at incentivizing the desired behavior?”) questions. We phased this by module: first they evaluated the grounded preference, then the staged plan conditioned on the grounded preference, then the reward code conditioned on the staged plan.
> >
> > > In 372 to 377, what are these scores that are being given? Is it the average satisfaction score across several categories? Given that the performance degrades quite a bit when you paraphrase the prompts, why do the authors claim that "the specific prompt matters less than the conceptual structure"?
> >
> > Thank you for the question. The analysis in Lines 372–377 serves two purposes: 1) to examine whether ROSETTA is robust to different versions of the prompt, 2) and to test whether the method still works when replacing proprietary models with fully open-source ones. The reported numbers are not human satisfaction scores; for this ablation, we did not have the annotator capacity to run additional satisfaction studies. Instead, the metric is the best reward achieved by an RL policy trained on the generated reward function (scaled to 0–1), which reflects the quality of the learned reward rather than human ratings.
> >
> > When we paraphrased the prompts using GPT-4o and Gemini 2.5, the resulting best-reward values remained within a narrow range (0.622 ± 0.030), indicating that the method is relatively insensitive to surface-level prompt wording and that the conceptual structure of ROSETTA is what drives performance. In addition, replacing the proprietary models with open-source ones led to even higher best-reward performance, showing that ROSETTA does not rely on proprietary systems and generalizes effectively across model families.
> >
> > > Why does the Sat. Score fall off so much after the first round?
> >
> > This is a great question! Some of this may be due to sample randomness, as given the nature of responding to open-ended preference, we could only collect a few hundreds of pieces of data. Other aspects: 1) in later rounds, annotators are thinking of less obvious preferences to give. 2) starting with the second round, preferences compound and become more complex, leading to minor deviations from the annotator’s goal.
> >
> > > In the real robot experiments, are the "two-iteration preference chains" generated by human annotators? What is the state dictionary that must be populated there?
> >
> > Yes, they are generated by human annotators! The state dictionary is the same as in ManiSkill’s default state representation excepting object orientation: object name mapped to 3D pose, gripper pose, gripper orientation, gripper open/close state.
> >
> > > How much did this all cost, particularly the annotations? This is relevant information for those who might choose to build on this paper.
> >
> > We agree this is of crucial importance!
> > - Generations: please see section G for cost analysis, including comparison to baselines. Please let us know if it would help to refer to this earlier in the paper. Each generation takes 300k-700k tokens and 3-5 minutes.
> > - Annotations: we paid each annotator $20 each time they gave eight new pieces of data, whether these were evaluations or preferences. We note that with optimistic estimations of time spent (1 minute watching the video a few times and thinking, 3 minutes reading and filling out the form), this is fairly generous. We chose a high payment due to 1) the unusual nature of the work, 2) the initial learning curve and time spent discussing with us, 3) the need to be on-call. Others wanting to use this approach could likely pay less without consequence. We have added this information to section D.1.

---

### Meta-Review · Area_Chair_9fHk · 2026-01-06

**Summary:**

This work extends a recent line of work where LLMs are used to generate reward functions in the form of code. It extends prior work (Eureka, Text2Reward) by modeling human preferences in its reward code generation. These are more difficult to exactly describe in code, since human preferences may reflect ambiguous or changing goals. The reviews were overall borderline: 4466. The reviewers generally agreed that the experimental results were solid and the paper well written. The main concerns fell into two categories: lack of details concerning the method and experiments, and lack of generalizability of the method beyond the considered benchmark (ManiSkill), which operates on low level proprioceptive inputs. In their response, the authors updated the paper with a number of clarifying details, and answered the reviewers with references to their paper containing the requested information, which I expect would have led to some increases in scores. Overall, I think this is a borderline paper since the problem setting is a bit niche/narrow, but it seems well executed within that scope. Therefore, I am still recommending except.

**Reviewer Concerns:**

Most of the clarification questions were addressed by the rebuttal, either in the form of paper updates with the requested details or pointers to the existing information in the paper. The questions about generalizability remain for the most part.

**Reviewer Scores:**

BmrL: 4 -> 6

Tpkx: 4 -> 6

zcpx: 6 -> 6

9E3U: 6 -> 6

---

### Decision · Program_Chairs · 2026-01-26

Accept (Poster)